# The impact of measures to reduce ambient air $PM_{10}$ concentrations originating from road dust, evaluated for a street canyon in Helsinki

Ana Stojiljkovic[1], Mari Kauhaniemi[2], Jaakko Kukkonen[2], Kaarle Kupiainen[1], Ari Karppinen[2], Bruce Rolstad Denby[3], Anu Kousa[4], Jarkko V. Niemi[4], Matthias Ketzel[5]

[1]Finnish Environment Institute (SYKE), Helsinki, P.O.Box 140, FI-00251, Helsinki, Finland
[2]Finnish Meteorological Institute, Helsinki, P.O. Box 503, FI-00101, Helsinki
[3]Norwegian Meteorological Institute, P.O. Box 43, Blindern, NO-0313 Oslo, Norway
[4]Helsinki Region Environmental Services Authority, P.O.Box 100, FI-00066, Helsinki, Finland
[5]Department of Environmental Science, Aarhus University, P.O. Box 358, DK-4000, Roskilde, Denmark

*Correspondence to*: Ana Stojiljkovic (*ana.stojiljkovic@ymparisto.fi*)

**Abstract**. We have evaluated numerically how effective selected potential measures would be for reducing impact of road dust on ambient air particulate matter ($PM_{10}$). The selected measures included reduction of the use of studded tyres in light-duty vehicles and reduction of the use of salt or sand for traction control. We have evaluated these measures for a street canyon located in central Helsinki for four years (2007-2009 and 2014). Air quality measurements were conducted in the street canyon for two years, 2009 and 2014. Two road dust emission models, NORTRIP and FORE, were applied in combination with the street canyon dispersion model OSPM to compute the street increments of $PM_{10}$ (i.e. fraction of $PM_{10}$ concentration originated from traffic emissions at the street level) within the street canyon. The predicted concentrations were compared with the air quality measurements. Both road dust emission models reproduced fairly well seasonal variability of the $PM_{10}$ concentrations, but under-predicted the annual mean values. It was found that the largest reductions of concentrations could potentially be achieved by reducing the fraction of vehicles that use studded tyres. For instance, a 30 % decrease in the number of vehicles using studded tyres would result in an average decrease of the non-exhaust street increment of $PM_{10}$ from 10 to 22 %, depending on the model used and the year considered. Modelled contributions of traction sand and salt to the annual mean non-exhaust street increment of $PM_{10}$ ranged from 4% to 20% for the traction sand, and from 0.1 % to 4 % for the traction salt. The results presented here can be used to support development of optimal strategies for reducing the high springtime particulate matter concentrations originated from the road dust.

## 1 Introduction

During the last couple of decades, strict regulations and technological innovations have led to a significant decrease of exhaust particulate emissions from road traffic. However, at the same time the decreases of non-exhaust traffic emissions have been much more moderate or even negligible, partly caused by the fact that these emissions have remained mostly unregulated (e.g., Kukkonen et al., 2018). Estimated relative contribution of non-exhaust emissions to the emissions of $PM_{10}$ from road transport increased from 30 % in 2000 to 60 % in 2016 (EEA, 2018).

The non-exhaust emissions of respirable particles, $PM_{10}$, include particles formed due to the wear of road
surface, brakes and tyres, and the suspension of particles that have been accumulated on the road surface and are
commonly referred to as road dust. The latter category is originated from (i) the wear of the road surface and the
tyres, (ii) traction control materials (sand and salt) and (iii) a range of other miscellaneous sources, such as the
deposited material from, e.g., road and building construction sites or surrounding environment, and the
deposition of materials to the surface from ambient air.
In northern European countries, the non-exhaust emissions have been one of the most important causes of high
ambient air $PM_{10}$ concentrations for several decades (e.g., Kukkonen et al., 1999, 2018; Kauhaniemi et al.,
2014). These have also resulted in exceedances of the daily $PM_{10}$ limit values set by the European Union (no
more than 35 occurrences of daily mean values exceeding 50 µg m$^{-3}$), especially during spring. In brief, the
mechanisms leading to such exceedances are (i) the accumulation of road dust on the road surfaces in winter, (ii)
the melting of snow and ice in spring, and (iii) the release of substantial amounts of suspended dust to the
atmosphere from the road surfaces during dry periods.
In the Nordic countries, it is necessary to use traction control measures (winter tyres, traction sanding and
salting) during the colder seasons to ensure traffic safety in snowy and icy weather.
The road wear associated with the use of studded winter tyres has been found to be the most significant source of
road dust (Kupiainen, 2007; Denby et al., 2013a; Kupiainen et al., 2016) that contributes to the high $PM_{10}$
concentrations. The use of traction sanding and salting contribute to a lesser degree to the amount of suspended
street dust; however, also these contributions may be significant (Denby et al., 2013a; Kupiainen et al., 2016).
Salt is commonly the preferred of the two traction control materials, but sanding has to be used in specific
weather conditions. These include in particular the conditions, for which the ambient temperatures are below -5
°C. Salting would then result in the freezing of the salt-water solution, and would not contribute to improving the
friction between the tyres and the street surface. The traction sand can directly contribute to the suspendable road
dust, if it contains particulate material that has specific grain sizes. There are also other processes, by which
traction sand can contribute: (i) via crushing of larger sand grains into smaller particles due to the passage of
tyre, and (ii) via abrasion of road surface by the contact of crushed stone and sand, and the tyres of passing
vehicles. The latter is commonly called the sandpaper effect (Kupiainen, 2007). According to Denby et al.
(2016), approximately 0.5 % of the total salt distributed on the roads can be released to the air as $PM_{10}$. As
approximately 200 000 tons of salt is spread out every year on the roads and streets in Finland, road salt can be a
significant source of the elevated $PM_{10}$ concentrations.
For the design of successful mitigation strategies for road dust, it would be valuable to assess contributions of
different sources to the $PM_{10}$ concentrations. Then it would also be possible to evaluate the efficiency and
impacts of potential abatement measures. Various modelling tools have been developed to facilitate such
analyses.
The aim of this study is to evaluate the effectiveness of selected potential measures for reducing the emissions
and concentrations of $PM_{10}$ originated from road dust. These measures include the reduction of the use of
studded tyres and the minimization of traction control material use. We have evaluated the effects of these
measures for a street canyon location in central Helsinki, for four years (2007-2009 and 2014). We have also
compared the predictions of the modelling system with the measured concentrations in the street canyon for two
years, 2009 and 2014. The non-exhaust $PM_{10}$ emissions associated with vehicular traffic were computed using
the road dust emission models NORTRIP (Denby et al., 2013a, 2013b) and FORE (Kauhaniemi et al., 2011).
Both emission estimates were then implemented in the OSPM street canyon dispersion model (Berkowicz, 2000)
to simulate the concentrations of $PM_{10}$ at the street level.

## 2 Materials and methods

### 2.1 Measurements

### 2.1.1 Study site description

The study was carried out for a segment of a major street called Hämeentie, located in central Helsinki. The
street segment is extending from south-west to north-east (at an angle of 56 degrees clockwise from the north).
The building block that surrounds the air quality measurement site extends over a distance of 91 m. The air
quality measurement site was at distances of 56 m and 35 m from the nearest junctions to the north-east and to
the south-west, respectively. The average height of the surrounding buildings in the studied segment of the street
is 26 m. The location of the study site in the city, and the applied meteorological and air quality stations are
presented in Fig 1a. The location of buildings and park areas in the immediate vicinity of this street segment is
presented in Fig. 1b. There is an open area and a small park to the north-east of the measurement site at distances
of approximately 60 and 200 m, respectively. There are several high trees in those areas that were estimated to
be approximately 10 m high. The street canyon is 32 m wide and it contains four lanes, two to both directions.

### 2.1.2 Traffic data and the use of studded tyres

The traffic volume data and, weekly and monthly variations of the traffic volume were based on the estimations
made by the Helsinki City Planning Department. The measured annual average weekday traffic volume is
available for 2015 for Hämeentie, and for 2007, 2008, 2009, and 2014 for a street that is a continuation street of
Hämeentie, located 600 m south-west from the site, called Pitkäsilta. Annual average weekday traffic volume
measured at Hämeentie in 2015 was adopted for year 2014. For other considered years, we have used measured
traffic volumes at Pitkäsilta, scaled by the ratio of annual average weekday traffic volumes at Hämeentie in 2015
and at Pitkäsilta in 2014.
The average hourly weekday daytime vehicle speeds are based on the values measured during the monitoring
campaigns in Hämeentie in 2007, 2009 and 2011. Measured values for 2007 and 2011 were adopted for years
2008 and 2014, respectively. The vehicle speeds for the night-time hours and weekend days were evaluated
using the measured diurnal and weekly cycles of vehicle speeds in Runeberginkatu (located 2 km southwest
from Hämeentie) in 2004. The traffic data for Hämeentie for years 2007-2009 and 2014 are summarized in Table
1. The average speeds of vehicles are clearly below the speed limit value (40 km h$^{-1}$), due to several junctions
and frequently occurring traffic congestion. This street is one of the major routes for public transport to the
centre of the city; the proportion of heavy-duty vehicles is therefore high, ranging annually from 29 to 30 %.
The use of winter tyres (studded or non-studded) is mandatory from December to February (inclusive) by
legislation in Finland. The studded tyres are allowed from November until the last day of March, or until
Monday one week after Easter, if it falls on a later date. However, studded tyres can be used also outside of this
period, if required by the weather conditions. Studded tyres are used only on light-duty vehicles. The maximum
annual share of light-duty vehicles using studded tyres during the study period was 80 %. For year 2014, the

transition from summer to winter tyres is based on the weekly counting of the vehicles with studded tyres in Helsinki (REDUST, 2014). For other considered years (2007-2009), such detailed information was not available, and the winter tyre season was therefore set to last from 23 October until 30 April. The transition between winter and summer tyres is assumed to be linear over a one-month period at the beginning and at the end of the winter tyre season.

### 2.1.3 Meteorological data

The meteorological data were obtained from two weather stations located at Kaisaniemi and Kumpula (Fig.1a) at distances of 1.0 and 2.4 km from the Hämeentie site, respectively. The data includes ambient temperature, relative humidity, precipitation, wind speed, wind direction, total cloud cover and global radiation. The monthly mean temperature and total precipitation values for the study period are presented in Fig. 2. In terms of the meteorological conditions relevant for the suspension emissions and dispersion conditions, all the years addressed in this study can be considered to be typical for this climate zone.

Using meteorological data at two urban stations could result in reduced representatives of the micrometeorological processes. Particularly, small-scale rain showers could be detected at the urban meteorological stations, but not at the study site, or the other way around.

### 2.1.4 Road maintenance data

The total number of relevant road maintenance activities is presented in Table 2 for different years during the study period. The salting and sanding events are the most and the second most frequent ones, respectively. Street cleaning is commonly done only once per year. The approximate seasonal timing of these activities is presented in Fig. 3. The complete data for the period from October to December was available only for year 2008. Most of the traction control measures (i.e., sanding and salting) have been done in winter and early spring, from January to March. Dust binding has been done mostly in spring, during March and April.

The information on the timing of road maintenance activities was available within an accuracy of six hours. The estimated dry masses of sand, traction salt (NaCl) and dust binding salt ($CaCl_2$) per application were 100 g m$^{-2}$, 10 g m$^{-2}$ and 6 g m$^{-2}$, respectively.

### 2.1.5 Air quality measurements

Kerbside air quality measurements were conducted in Hämeentie in 2009 and 2014. Urban background air quality measurements were made at the station of Kallio, which is located at a distance of 700 m north-west from the Hämeentie site.

### 2.2 Models

### 2.2.1 The models for evaluating the suspension emissions

The non-exhaust $PM_{10}$ emissions for 2007-2009 and 2014 were computed using the NORTRIP and FORE models. A brief overview of the models' structure and their application in this study is presented in this section. The reader is referred to Denby et al. (2013a, 2013b) (NORTRIP) and Kauhaniemi et al. (2011, 2014) (FORE) for comprehensive description of the models and parameter definitions.

**The road dust emission model NORTRIP**
The NORTRIP model (NOn-exhaust Road TRaffic Induced Particle emissions) is described in Denby et al.
(2013a, 2013b) and comprises two sub-models that describe the road dust and surface moisture mass balance.
Coupled they are used to predict emission of the road dust, which results from the direct emissions of vehicle
related wear (road, brakes and tyre) and suspension of wear products, salt and sand accumulated on the road
surface.
The road dust emission calculation is based on the total wear rates and the size distributions of the different wear
sources. The basis road wear rate for studded tyres is determined using the Swedish road wear model (Jacobson
and Wågberg, 2007) and can be adjusted for different road surface types. The basis brake and tyre wear rates and
size distributions used in this study are taken from Boulter (2005). The suspension of road dust induced by
passing vehicles is accounted for in the NORTRIP model using a suspension factor. The suspension factor in
NORTRIP was initially derived by optimising the model against ambient air measurements that clearly show the
decay in PM emissions at the end of the studded tyre season and is described in Denby et al. (2013a).
Application of the model to many datasets since then does not indicate the need for significant changes to this
suspension factor.
Table 3 shows parameters relevant for calculation of emissions from wear and suspension for light-duty vehicles
that are valid for vehicle speeds of 70 km $h^{-1}$ for wear, and 50 km $h^{-1}$ for $PM_{10}$ fraction and suspension. These
speeds are significantly higher than speeds encountered in this study, and the values are presented for reference
only. The road wear and suspension are considered to be linearly dependent on vehicle speed. The wear and
suspension rates for the heavy-duty vehicles are assumed to be 5 and 10 times larger than those for light-duty
vehicles, respectively.
The surface moisture, as calculated by the surface moisture model, determines the suspension and retention of
the road dust and salt. The surface moisture is a product of precipitation, condensation and wetting whereas the
removal of surface moisture happens through drainage, evaporation and spray. Additionally, drainage and spray
will contribute to removal of dust and salt from the road surface. The energy balance model is used to predict
condensation and evaporation from the road surface.
The NORTRIP model input data includes information on street configurations, traffic data (traffic volume and
composition, vehicle speed and tyre type), meteorological data (solid and liquid precipitation, wind speed,
temperature, radiation, cloud cover, and humidity) and road maintenance activity data.
Road maintenance activities included in the NORTRIP model are traction salting and sanding, dust binding,
cleaning and ploughing. Traction sand directly contributes to the suspendable road dust mass, depending on its
particle size distribution. Size distribution measurements of traction sand used in the Helsinki Metropolitan Area
showed that 0.4-2.5% of the sanding material is in the suspendable fraction (defined as the size fraction < 200
μm) (Kulovuori et al., 2019). In this study, the amount of suspendable material in sand was set to be equal to
2%. Salt contributes directly to the dust loading, when not in solution, and impacts on the predicted surface
conditions via surface vapour pressure depression that reduces evaporation (Denby et al., 2013b). In the model,
cleaning and ploughing reduce the amount of road dust and salt on the road surface with a predefined efficiency.
The effect of street cleaning will depend on the method used and initial amount of road dust available on the
street surface (e.g. REDUST, 2014). In this study, assumed removal efficiency for cleaning and ploughing are set
to be 1% and 0.1% for the non-suspendable and suspendable fraction of the road dust, respectively.
The output of the model consists of hourly time series for the emissions from wear sources and from salt and
sand in the size fraction of $PM_{10}$.
**The road dust emission model FORE**
The FORE model (Forecasting Of Road dust Emissions) has been developed to evaluate the particulate matter
emissions from road and street surfaces. It is based on the particle suspension emission model developed by
Omstedt et al. (2005). The model considers emissions formed by the wear of road surface and from traction sand
and the suspension of road dust particles into the atmosphere. The model version does not address the emissions
caused by the wear of vehicle components (e.g. brake and tyre wear).
The use of the model requires as input hourly meteorological data (i.e., total precipitation, temperature, dew
point temperature, relative humidity, wind speed, radiation and cloud cover), the roughness length, the share of
studded tyres, and the dates of the street sanding.
The model uses empirical reference emission factors, which have different values depending on the time of the
year, the size fraction of particles ($PM_{10}$ or $PM_{2.5}$), and the traffic environment (urban or highway). The reference
emission factor will be higher for the time of the year when sanding and studded tyres are commonly used
(referred to as 'sanding period') compared to the rest of the year (referred to as 'non-sanding period').
We have adopted the reference emission factors evaluated for Stockholm estimated and further explained by
Omstedt et al. (2005); i.e., 1200 and 200 µg veh$^{-1}$ m$^{-1}$, for sanding (Oct-May) and non-sanding (Jun-Sep) period,
respectively. The climatic conditions, studded tyre shares and the procedures of using traction sand are fairly
similar in Stockholm and Helsinki, although the difference in used amounts of sand and salt can be significant.
The dust layer, which will be accumulated on the street surface during wet conditions, depends on the traction
sanding and the road wear. In the FORE model, equal contributions are assumed for the dust layers on the street,
originating from the road wear and from the traction sand. The dust layer is reduced by the suspension of
particles due to the air flow and by runoff due to precipitation.
The suspension of road dust particles in the air is controlled by road surface moisture that is based on modelling
of precipitation, runoff, and evaporation. The effect of terrain on wind is defined by roughness length which is
needed for the evaluation of the evaporation (Omstedt et al. 2005). In the present case, the roughness length was
derived from the average building height (26 m) in the studied street section. This resulted in the roughness
length value of 2.6 m.
The model does not allow for the dependencies of emissions on vehicle speed and fleet composition. In the
FORE model, we have used as input the studded tyre share of the whole traffic fleet of Hämeentie, including
both light-duty and heavy-duty vehicles. As studded tyres are only used in light-duty vehicles at the study site,
corresponding share of studded tyres in the total traffic fleet is relatively lower. For instance, assuming that 80%,
50%, 30% or 0% of the light-duty vehicles uses studded tyres, the studded tyre share of the whole traffic fleet is
approximately 57%, 35%, 21% and 0%, respectively.
**2.2.2 Evaluation of the vehicular exhaust emissions**
The exhaust emission factors were obtained from the LIPASTO emission modelling system (Mäkelä, 2015). The
LIPASTO emission factors are defined separately for five vehicle categories (personal cars, vans, buses, lorries
without a trailer, and lorries with a trailer). The dependencies of emission factors on the vehicle speeds were not

explicitly taken into account; however, they allow for urban driving conditions, i.e., traffic cycles that contain frequent accelerations, decelerations and idling. The vehicular exhaust emission factors for particulate matter used in this study are presented in Table 4. As expected, the emission factors were the largest for lorries equipped with a trailer. The emission factors are an order of magnitude larger for heavy-duty vehicles and vans, compared with the personal cars.

### 2.2.3 The street canyon dispersion model OSPM

The street canyon dispersion model OSPM is based on a combination of a Gaussian plume model and an empirical box model. For a detailed description of this model, the reader is referred to Berkowicz (2000). A brief overview of the model structure and its application in this study is presented here.

The OSPM model requires as input data information on the street configuration, hourly time series of the traffic data, the exhaust- and non-exhaust emissions, the meteorological parameters (wind speed and direction), and the urban background concentrations.

The input information on the street configuration includes the geometry of the studied street segment; introduced in Section 2.1.1. The ratio of canyon height (26 m) and width (32 m) gives an aspect ratio of 0.8. Thus, the studied street is considered as a wide street canyon. The aspect ratio of studied street is close to an ideal value in view of the performance of the OSPM model; the model was developed for street canyons with an aspect ratio close to unity.

We have also taken into account the geometries of nine street crossings and two parks that are outside of the studied street segment. These so-called exceptions on canyon walls need to be considered, although they are outside of the studied street segment, as they are situated less than 200 m from the receptor points. Otherwise, the OSPM model will assume that the row of buildings continues over a very large distance (Berkowicz et al., 2003). The geometries of street crossings and parks are considered in the model for various wind sectors and so-called building height exceptions.

Trees add the porosity of a street canyon, and thus have an influence on dispersion and deposition conditions. However, the OSPM model does not consider any obstacles in the street canyon.

The completeness of the meteorological and background concentration data used as input for the OSPM calculations was excellent. Average data coverage for wind speed and direction, and background concentrations was 98%.

Traffic induced turbulence depends in the model on traffic flow and composition (light and heavy vehicles), as well as on the traffic speed. The hourly average traffic volume and speed data were used as input separately for light-duty vehicles (i.e., passenger cars and vans) and heavy-duty vehicles (i.e., busses and lorries).

### 3. Results and discussion

Two road dust emission models, NORTRIP and FORE, were applied to compute the vehicular non-exhaust $PM_{10}$ emissions that were, together with the exhaust emissions, then used as input in the OSPM street canyon dispersion model to simulate street level $PM_{10}$ concentrations.

We have (i) compared predictions of the models to the measured $PM_{10}$ concentrations (Section 3.1.), (ii) evaluated key uncertainties in the road dust and dispersion modelling for the study site (Section 3.2.), and (iii)

simulated the effects of changes in studded tyre share and the impacts of traction sanding and salting on ambient
air $PM_{10}$ concentrations (Section 3.3).
For the comparison with the measured concentrations we have focused on the street increments of $PM_{10}$. The
measured and predicted street increments were obtained by subtracting the measured urban background
concentrations from the measured and predicted concentrations in the street canyon, respectively. Effects of
measures intended to reduce road dust emissions were subsequently estimated for the non-exhaust part of the
street increments, as a relative difference compared to reference case. Non-exhaust street increment is a fraction
of the modelled street increment $PM_{10}$ concertation that originates from the non-exhaust traffic induced particle
emissions. The results are presented as annual and seasonal mean values. Seasons are defined as follows: winter
(1 January to 14 March), spring (15 March to 31 May), summer (1 Jun to 30 September) and autumn (1 October
to 31 December).
### 3.1. Comparison of predicted and measured $PM_{10}$ concentrations
The kerbside air quality measurements in Hämeentie were performed in 2009 and 2014. The total observed
annual mean concentrations of $PM_{10}$ were 24 µg m$^{-3}$ and 23 µg m$^{-3}$ in 2009 and 2014, respectively, and were
slightly above the WHO guidelines (annual mean $PM_{10}$ concentration should not exceed 20 µg m$^{-3}$). The EU
daily limit value (50 µg m$^{-3}$) was exceeded on 16 days in 2009, and on 21 days in 2014 (Malkki et al. 2010;
Malkki and Loukkola 2015). Although the number of exceedances was below the allowed number of 35 days,
elevated $PM_{10}$ concentrations caused by the road dust in spring can cause adverse health impacts and reduce the
comfort of living. The urban background contribution to the concentrations measured at the street level in
Hämeentie was substantial, i.e., 64%, averaged over the two years with available data (2009 and 2014).
The time series of modelled and observed daily mean street increment concentrations of $PM_{10}$ for years 2009 and
2014 are presented in Fig. 4. The annual and seasonal mean values are presented in Table 5. In 2009, the
observed seasonal variation was more pronounced, compared with the corresponding results for 2014, as shown
both by the results in Fig. 4 and Table 5. The observed street increment in spring was clearly the highest for both
years, compared with that in the other seasons.
In 2009, a snow layer was formed on the street in the second half of January, and lasted until the end of March.
The month of April was warmer than average and with less precipitation. The observed daily mean $PM_{10}$
concentrations started to increase in the latter part of March and prevailed on a relatively high level until the end
of April. Night frosts postponed the street cleaning that commonly starts in March, to the beginning of April.
This contributed, together with the lack of precipitation, to the existence of a prolonged road dust season.
On the other hand, the winter of 2014 was milder than average. The snow cover lasted only for a short time
between January and February, and the thermal spring started early. The observed $PM_{10}$ concentrations were on
average substantially lower during spring, compared with those in 2009, caused by both early spring cleaning
procedures and fortunately timed precipitation events.
Both models can be considered to have reproduced the seasonal variability fairly well, but they under-predict the
annual mean values. The street increments of $PM_{10}$ predicted by the FORE model are higher than the observed
values in winter and lower in spring, for both years. The NORTRIP model systematically under-predicts the
measured concentrations. The NORTRIP model reproduced observed variation of the daily mean street
increment concentrations reasonably well with the coefficients of determination $R^2$ of 0.51 and 0.32 for 2009 and
2014, respectively. The corresponding correlations for the FORE model were slightly lower ($R^2 = 0.25$ and 0.20
for 2009 and 2014). The correlation of the hourly mean street increment concentrations, compared with the
corresponding values for the daily means, was substantially lower in case of the NORTRIP computations ($R^2 =$
0.38 and 0.25 for 2009 and 2014, respectively). This was probably due to the higher uncertainties in evaluating
the hourly variation of the street surface conditions. In case of the FORE model ($R^2 = 0.26$ and 0.20, for 2009
and 2014, respectively), the daily and hourly correlations were very similar to each other. Additional results of
the statistical analyses for the daily mean street increments of $PM_{10}$ are presented in Appendix A.
**3.2 Evaluation of the uncertainties of the modelling**
There are significant uncertainties in the modelling of the road dust and dispersion modelling associated to the
numerous model input values and parameters used for the model computations. Additionally, uncertainties that
can affect the accuracy of the whole modelling system are potentially missing road dust sources or source
categories. Such sources could be the migration of dust from adjoining streets, the off-road sources (such as
sidewalks and parking lots) and the traction sand used by trams.
We have analysed and numerically evaluated selected key uncertainties related to the application of the two road
dust emission models, and to the street canyon modelling for the Hämeentie site.
**3.2.1 Uncertainties of the NORTRIP model**
Denby et al. (2013b) previously studied extensively the sensitivity of the NORTRIP model to a wide range of
input parameters and demonstrated ability of the model to reproduce the mean concentrations of $PM_{10}$ within a
range of ± 35 % of observations for a number of data sets. However, the results of the present study were outside
of the above mentioned range of uncertainties.
The results presented in Section 3.1. show that the NORTRIP model systematically under-predict observed $PM_{10}$
concentrations for Hämeentie. Road wear particles created by the studded tyres dominate in the road dust
emissions. In the NORTRIP model, the wear rate caused by studded tyres depends on the properties of asphalt
road surface (such as stone sizes and wear resistance) and vehicle speed. In this study, we have used wear rates
derived for the reference road surface type (ABS16 with porphyry from Älvdalen) in the Swedish road wear
model (Jacobson and Wågberg, 2007) which is one of the most wear resistant road surfaces used in Sweden. The
wear rates in the Swedish road wear model are based on laboratory and field experiments and provide an average
under both prevailing dry and wet conditions. However, influence of surface moisture that increases the wear is
not directly considered in the model calculations. Denby et al. 2013a estimated the typical wear rates to be from
2 to 5 g $km^{-1}$ $veh^{-1}$ and acknowledged significantly variation of these values depending on the material used with
increased wear rates for roads with the poor quality surfaces. Hämeentie is paved with the stone matrix asphalt
but further detailed information about road surface parameters was not available, which is a source of uncertainty
in evaluating the studded tyre wear rates.
We found that numerically doubling the studded tyre wear rate would increase the mean street increment
concentrations of the $PM_{10}$ computed with the NORTRIP model by 34%. This would therefore result in model
predictions that would be in better agreement with the measurements. However, increasing wear rate will not
affect significantly coefficient of determination ($R^2$) because this parameter is a measure of correlation between
modelled and measured values not measure of bias of mean values.
The studded tyre wear rate is also assumed to be linearly dependent on vehicle speed (Denby et al., 2013a). In all
previous calculations using the NORTRIP model (Denby et al., 2013b), the vehicle speeds have been larger than
40 km h$^{-1}$. The dependency on vehicle speed may be non-linear for the lower traffic speeds encountered in this
study (< 30 km h$^{-1}$) due to congestion. The NORTRIP model also does not account for the influences of
congested driving conditions, in which acceleration and deceleration will likely result in an enhanced road wear.
In summary, it is possible that an underestimation of the studded tyre wear rate in congested low vehicle speed
conditions, for this particular road surface, could contribute to the under-predictions by the NORTRIP model.
### 3.2.2 Uncertainties of the FORE model
The key parameter in the FORE model is the reference emission factor, which sets a baseline value for the
predicted suspension emissions. In this study, we have used the reference emission factors estimated by Omstedt
et al. (2005) based on the measurements in Hornsgatan in Stockholm. Although the climatic conditions were
similar during the Hornsgatan measurement campaign and the present study, the different traffic conditions
could in principle have caused differences that will be reflected in the values of the baseline emissions.
We have therefore estimated numerically, how using the physically largest feasible values of the reference
emission values would increase the predictions of the FORE model. The base case PM$_{10}$ reference emission
factors for the sanding and non-sanding periods in Omstedt et al. (2005) were 1200 and 200 µg veh$^{-1}$ m$^{-1}$,
respectively. The assumed numerical cases used the higher PM$_{10}$ reference emission factors for the sanding and
non-sanding periods, i.e., 1500 and 300, and 3200 and 400, respectively. For the assumed numerical cases, the
annual mean street increment concentrations of PM$_{10}$ would increase from 23% to 118%.
The FORE model does not address the influences of salting, street cleaning and dust binding. The suspension
emissions are also, for simplicity, modelled for the whole vehicle fleet. This approach does not take into account
the details of the vehicle speeds and the composition of the vehicle fleet.
In summary, an under prediction of the baseline emissions could have contributed to the under-prediction of
suspended PM$_{10}$ concentrations found in this study. Neglecting the effects of salting, street cleaning and dust
binding could cause a reduced correlation of the measured and predicted concentration values.
### 3.2.3 Uncertainties of the OSPM model
The OSPM model contains the so-called roof parameter (fRoof), which is used to relate the measured or
modelled wind speed at a meteorological mast with the wind speed at roof level. The value of this parameter
depends on building and roughness situations around the meteorological station. In this study, we have used the
roof parameter value of 0.4, which is based on the model-measurement studies conducted in Copenhagen by
Ketzel et al (2012).
However, some other studies have suggested a higher value of 0.6 (OSPM FAQ, 28.03.2017). The numerical
computations showed that using the higher value of the fRoof parameter would not improve agreement between
model predictions and measurements for Hämeentie site. The mean street increment PM$_{10}$ concentration over the
two years (2009 and 2014) was approximately 26% lower, using this higher value of the roof parameter,
compared to that with fRoof value of 0.4.
**3.3 Impact of the reductions in studded tyre use and road maintenance measures**
We have assessed numerically the impact of changes in selected traction control measures on the non-exhaust
street increments of $PM_{10}$.The selected numerical cases are presented in Table 6. In the so-called reference case,
we have assumed that all reported road maintenance activities have been done, and the maximum share of the
light-duty vehicles using studded tyres is equal to the observed value. The maximum observed share of vehicles
using studded tyres (80%) was numerically reduced to 50% (ST 50%), 30% (ST 30%) and 0% (no ST). We also
assumed that all recorded sanding and salting events would not have been done in 'no Sand' and 'no Salt' case,
respectively. Both road dust emission models (NORTRIP and FORE) were applied to assess the impacts of the
reduced fraction of studded tyres and the impact of traction sanding. The impact of traction salt was studied only
using the NORTRIP model.
The computed changes in the modelled non-exhaust increments of $PM_{10}$, relative to the reference case are
presented in Fig. 5. The largest reductions of concentrations can be achieved by reducing the use of studded tyres
in favour of the non-studded winter tyres. For the most extreme case with no studded tyres in traffic, the average
decreases in the non-exhaust street increments of $PM_{10}$ over four year period were 39% and 40% for the
NORTRIP and FORE model, respectively. In case where the reference maximum studded tyre share was reduced
by 30%, average decreases in modelled annual non-exhaust street increments of $PM_{10}$ were 16% (NORTRIP) or
17% (FORE). Varying effect of the same studded tyre reduction between different years can be attributed to the
changing meteorological conditions that influence suspension emission and road dust removal processes as well
as the dispersion conditions.
The impact of studded tyre reductions can be further enhanced by improving the quality of road surfaces. Larger
aggregate sizes that are made from rocks more resistant to wear in the asphalt roads, or the use of alternative
road surfaces can reduce $PM_{10}$ emissions (Gustafsson et al. 2009; Gustafsson and Johansson 2012).
The number of reported sanding events in Hämeentie was 9 in 2007 and 18 in 2009 and 2014 (Table 2). In year
2008, all traction control was done by salting. All sanding events occurred during January and February. Salting
was extensively used between January and March during the study period with 17 to 49 salting events per year.
The results for the 'no Sand' and 'no Salt' cases give an indication of the overall contribution of implemented
sanding and salting to the non-exhaust street increments of $PM_{10}$ in Hämeentie. Without taking into account
reported sanding events, both road dust emission models predict similar changes in the modelled street increment
concentrations averaged over the four years; however, with different seasonal variation. The modelled
contribution of sanding to the annual mean non-exhaust street increment of $PM_{10}$ ranges from 4 to 20%,
depending on the year and the model considered. The NORTRIP model predicts highest impact of sanding in
spring months and indicates that sanding influence extends throughout summer. The impact of sanding predicted
by the FORE model is limited to winter, spring, and autumn owing to model's concept regarding the sanding
implementation.
The traction salt is efficiently removed from the street surfaces by drainage and vehicle spray processes, which
are affected by precipitation (Denby et al., 2016). In dry conditions, traction salt can significantly contribute to
the $PM_{10}$ concentrations. The predicted change in annual mean non-exhaust street increments of $PM_{10}$ after
exclusion of reported salting events ranges from -0.1% to -4%.
The results demonstrate that traction sanding and salting are potentially significant sources of the road dust.
However, immediate restrictions in their use could jeopardize traffic fluency and safety. Optimizing spatially and

temporally the use of traction control materials can reduce the impacts of road dust on the $PM_{10}$ concentrations. The impact of traction sand on suspended road dust will depend on the frequency of the sanding operations, and the amount and quality of sanding material. The use of sanding material with high resistance to fragmentation and with removed fine particulate fractions will reduce the contribution of sanding to the suspendable road dust (Tervahattu et al., 2006). From an air quality perspective, substituting sand for less dust forming materials, such as salt, would be beneficial. However, this may not be always possible, due to the prevailing weather conditions, and also in areas, which need to be protected from the negative environmental effects of the conventional traction salt, sodium chloride (NaCl). Alternatives to sodium chloride, such as other chlorine based salts and organic salts, have been tested for use in sensitive groundwater areas in Finland (e.g. Hellstén et al., 2001, 2002); however, their widespread use has not been introduced.

## 4 Conclusions

We have conducted numerical computations regarding the effectiveness of potential measures to reduce impact of road dust on ambient air $PM_{10}$ concentrations. The selected measures included reduction of the use of studded tyres in light-duty vehicles and reduction of the use of traction sanding and salting. The effects of these measures were analysed for a street canyon in central Helsinki. Two road dust emission models, NORTRIP and FORE, were used in combination with the street canyon dispersion model OSPM. We have compared predictions of the modelling system with the available street canyon measurements for a period of two years and evaluated variability and uncertainties associated with various modelling approaches. Impact of selected traction control measures was estimated for the non-exhaust street increments of $PM_{10}$.

The NORTRIP model is a process based model that describes wear processes, traffic induced suspension of accumulated road dust and impact of road maintenance activities (salting, sanding, dust binding, cleaning and ploughing) on both dust load and road surface moisture. It includes dependences on vehicle speed, tyre type, vehicle category (light and heavy-duty vehicles) and road surface properties that enable a comprehensive evaluation of the road dust abatement measures. However, the model requires extensive input data that may not be available (such as, e.g. road maintenance data and the properties of the road surface). This may present a challenge in application of the NORTRIP model. On the other hand, the FORE model requires relatively much less input data. However, it relies on the reference emission factors, which need to be computed based on local air quality measurements. The FORE model considers two road dust sources, viz. road wear and traction sand. The model takes into account neither the dependence of emissions on vehicle speed and traffic fleet composition, nor the influence of traction salting and dust control measures (i.e., dust binding and street cleaning). These factors limit the application of the FORE model for evaluation of a wider range of measures to reduce road dust.

Both road dust emission models reproduced the seasonal variability of the concentrations of $PM_{10}$ fairly well, but under-predicted the annual mean values. The street increments of $PM_{10}$ predicted by the FORE model tended to be higher than the observed values in winter and lower in spring, whereas the NORTRIP model systematically somewhat under-predicted the measured concentrations. The daily mean street increment concentrations predicted by NORTRIP correlated reasonably well with the measured values; the correlation was better than the corresponding one for the FORE model. An underestimation of the studded tyre wear rate in congested low vehicle speed conditions, which are common for the Hämeentie site, could contribute to the under-predictions by

the NORTRIP model. In case of the FORE model, the main uncertainties were the underestimation of the baseline emission factor and neglecting the effect of salting, street cleaning and dust binding.

There are substantial differences in the structure and mathematical treatments of various processes in the NORTRIP and FORE models. Despite the differences, these models predicted a very similar distribution of changes in the $PM_{10}$ concentrations for the studied cases.

The results demonstrate that changes in the current traction control measures can significantly reduce the impact of road dust on ambient air $PM_{10}$ concentrations. The largest reductions in $PM_{10}$ concentrations can be achieved by reducing studded tyre use in favour of the non-studded winter tyres. For instance, in case where the reference maximum studded tyre share was reduced by 50 %, average decrease in the non-exhaust street increment of $PM_{10}$ was from 16 % to 34 %, depending on the model used and the year considered. However, the effectiveness of the studded tyre reductions is also dependent on other factors, such as the quality of the road surfaces, vehicle speed and vehicle driving cycles. In addition, both the fluency and safety of vehicular traffic and the implementation of street maintenance measures are substantial economic issues. The reduction of the use of studded tyres would be beneficial also due to the reduced costs for the repairing of road surfaces.

Modelled contribution of traction sanding to the annual mean non-exhaust street increment of $PM_{10}$ during the study period ranged from 4 % to 20 %. The impact of traction salting was estimated using only the NORTRIP model. Completely removing street salting reduced the non-exhaust street increment of $PM_{10}$ from 1% to 4% on annual level.

Based on the results, optimizing the use of traction control materials can reduce impact of road dust on $PM_{10}$ concentrations. For example, substituting sanding for a less dust forming materials such as salt, whenever possible, would reduce the amount of road dust, but this measure would not completely eliminate road dust emissions. Additionally, the contribution of sanding can further be reduced by choosing the sand materials that are wear resistant and do not contain the finer grain fractions.

We have demonstrated that there is a substantial potential for reducing the impact of road dust on ambient air $PM_{10}$ concentrations, by changing the traction control measures of both vehicles (studded tyre use) and street and road maintenance (sanding and salting). The results presented here can be used to support the development of feasible strategies for reducing the high springtime particulate matter concentrations originated from the road dust.

**Appendix A: Results of the statistical analyses for the daily mean street increments of $PM_{10}$ for Hämeentie**
**in 2009 and 2014.**
Table A1 presents the statistical values for daily mean street increment $PM_{10}$ concentrations for 2009 and 2014
calculated on annual and seasonal level. The error of both models is lowest during summer and highest for
winter (FORE) or spring (NORTRIP). The RMSE indicates substantial inaccuracies in daily $PM_{10}$ street
increment concentrations for both models.
Table A1. Statistical values for modelled daily mean street increment of $PM_{10}$ for the NORTRIP and FORE
models for 2009 and 2014, calculated on annual and seasonal level.

| NORTRIP 2009 | | | | | | |
|---|---|---|---|---|---|---|
| **Statistical parameter** | | **Winter** | **Spring** | **Summer** | **Autumn** | **Annual** |
| RMSE | Root mean square error | 8.4 | 15.4 | 3.1 | 5.5 | 8.7 |
| IA | Index of agreement | 0.50 | 0.62 | 0.72 | 0.49 | 0.67 |
| F2 | Factor-of-two | 54 | 42 | 75 | 69 | 62 |
| $R^2$ | Coefficient of determination | 0.06 | 0.61 | 0.56 | 0.15 | 0.51 |
| FB | Fractional bias | -0.44 | -0.74 | -0.41 | -0.50 | -0.57 |
| AvgCp | Average of predicted data | 4.6 | 9.4 | 4.4 | 3.7 | 5.3 |
| AvgCo | Average of observed data | 7.2 | 20.5 | 6.7 | 6.1 | 9.6 |
| N | Number of data points | 71 | 78 | 122 | 89 | 360 |
| **FORE 2009** | | | | | | |
| **Statistical parameter** | | **Winter** | **Spring** | **Summer** | **Autumn** | **Annual** |
| RMSE | Root mean square error | 13.7 | 13.4 | 2.7 | 4.7 | 9.2 |
| IA | Index of agreement | 0.21 | 0.70 | 0.78 | 0.63 | 0.67 |
| F2 | Factor-of-two | 42 | 56 | 80 | 69 | 64 |
| $R^2$ | Coefficient of determination | 0.00 | 0.52 | 0.43 | 0.23 | 0.25 |
| FB | Fractional bias | 0.55 | -0.49 | -0.20 | -0.08 | -0.13 |
| AvgCp | Average of predicted data | 12.7 | 12.4 | 5.5 | 5.6 | 8.4 |
| AvgCo | Average of observed data | 7.2 | 20.5 | 6.7 | 6.1 | 9.6 |
| N | Number of data points | 71 | 78 | 122 | 89 | 360 |
| **NORTRIP 2014** | | | | | | |
| **Statistical parameter** | | **Winter** | **Spring** | **Summer** | **Autumn** | **Annual** |
| RMSE | Root mean square error | 9.6 | 10.6 | 4.1 | 9.7 | 8.5 |
| IA | Index of agreement | 0.47 | 0.62 | 0.63 | 0.47 | 0.58 |
| F2 | Factor-of-two | 45 | 44 | 62 | 25 | 45 |
| $R^2$ | Coefficient of determination | 0.29 | 0.44 | 0.44 | 0.10 | 0.32 |
| FB | Fractional bias | -1.11 | -0.76 | -0.54 | -1.17 | -0.83 |
| AvgCp | Average of predicted data | 2.2 | 6.9 | 4.4 | 2.2 | 3.9 |
| AvgCo | Average of observed data | 7.7 | 15.3 | 7.6 | 8.5 | 9.5 |
| N | Number of data points | 73 | 78 | 122 | 92 | 365 |
| **FORE 2014** | | | | | | |
| **Statistical parameter** | | **Winter** | **Spring** | **Summer** | **Autumn** | **Annual** |
| RMSE | Root mean square error | 10.5 | 8.3 | 3.8 | 8.7 | 7.8 |
| IA | Index of agreement | 0.42 | 0.74 | 0.66 | 0.48 | 0.64 |
| F2 | Factor-of-two | 36 | 62 | 68 | 49 | 55 |
| $R^2$ | Coefficient of determination | 0.02 | 0.41 | 0.32 | 0.15 | 0.20 |
| FB | Fractional bias | 0.16 | -0.36 | -0.39 | -0.80 | -0.34 |

| | | | | | | |
|---|---|---|---|---|---|---|
| AvgCp | Average of predicted data | 9.0 | 10.7 | 5.1 | 3.6 | 6.7 |
| AvgCo | Average of observed data | 7.7 | 15.3 | 7.6 | 8.5 | 9.5 |
| N | Number of data points | 73 | 78 | 122 | 92 | 365 |

*Author contributions*. AS, MK[2], JK and KK designed this study. AS performed NORTRIP model calculation, numerical analyses and wrote a first draft of the paper. KH performed FORE and OSPM model calculations. JK and MK[2] significantly contributed in revision of the manuscript. AK[4] and JVN provided supporting air quality data. KK, BRD, AK[2] and MK[5] provided professional comments to improve the manuscript.

### Acknowledgements

This study has been a part of the research projects "Non-exhaust road traffic induced particle emissions - Evaluating the effect of mitigation measures on air quality and exposure (NORTRIP-2)", funded by the Nordic Council of Ministers (Project number KOL-1408), "Understanding the link between Air pollution and Distribution of related Health Impacts and Welfare in the Nordic countries (NordicWelfAir)", funded by Nordforsk, (Project #75007) and "Global health risks related to atmospheric composition and weather (GLORIA)" and "Environmental impact assessment of airborne particulate matter: the effects of abatement and management strategies (BATMAN)", funded by the Academy of Finland.

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

Table 1. Summary of traffic data at Hämeentie for four years.

| Year | Annual average daily traffic (vehicles day$^{-1}$) | Share of heavy-duty vehicles (%) | Mean speed (km h$^{-1}$) |
|---|---|---|---|
| 2007 | 11400 | 29 | 27 |
| 2008 | 9700 | 29 | 27 |
| 2009 | 10110 | 29 | 27 |
| 2014 | 9050 | 30 | 25 |

Table 2. The number of road maintenance measures in Hämeentie for four years. Number of ploughing events
was computed using the NORTRIP model.

| Year | Sanding events | Traction salting (NaCl) | Dust binding (CaCl$_2$) | Street cleaning | Ploughing |
|---|---|---|---|---|---|
| 2007 | 9 | 21 | 1 | 2 | 7 |
| 2008 | 0 | 49 | 4 | 1 | 14 |
| 2009 | 18 | 40 | 3 | 1 | 19 |
| 2014 | 18 | 17 | 10 | 1 | 9 |

Table 3. The wear and suspension rates for the light-duty vehicles and the fraction of wear material in the size
range of PM$_{10}$ used in the NORTRIP model. The reference speed is 70 km h$^{-1}$ for wear and 50 km h$^{-1}$ for PM$_{10}$
fraction and suspension.

| | Studded tyres | Winter tyres | Summer tyres | PM$_{10}$ fraction (%) |
|---|---|---|---|---|
| Road wear (g km$^{-1}$ veh$^{-1}$) | 2.88 | 0.15 | 0.15 | 28 |
| Tyre wear (g km$^{-1}$ veh$^{-1}$) | 0.1 | 0.1 | 0.1 | 10 |
| Brake wear (g km$^{-1}$ veh$^{-1}$) | 0.01 | 0.01 | 0.01 | 80 |
| Road dust suspension rate (veh$^{-1}$) | 5.0x10$^{-6}$ | 5.0 x10$^{-6}$ | 5.0 x10$^{-6}$ | - |

Table 4. The vehicular exhaust particulate matter emission factors (g km$^{-1}$ veh$^{-1}$) for four years, based on the
LIPASTO emission modelling system.

| Vehicle type | 2007 | 2008 | 2009 | 2014 |
|---|---|---|---|---|
| Personal cars | 0.03 | 0.03 | 0.02 | 0.01 |
| Vans | 0.15 | 0.14 | 0.14 | 0.10 |
| Buses | 0.29 | 0.25 | 0.21 | 0.12 |
| Lorries, no trailer | 0.19 | 0.16 | 0.13 | 0.09 |
| Lorries with trailer | 0.55 | 0.47 | 0.35 | 0.23 |

Table 5. Annual and seasonal mean observed and modelled street increments of $PM_{10}$ ($\mu g\ m^{-3}$) for Hämeentie in 2009 and 2014.

| Year | | Winter | Spring | Summer | Autumn | Annual mean |
|---|---|---|---|---|---|---|
| 2009 | Observed | 7.8 | 20.1 | 6.9 | 6.4 | 10.1 |
| | NORTRIP | 5.3 | 9.4 | 4.5 | 3.9 | 5.7 |
| | FORE | 13.4 | 12.4 | 5.6 | 6.0 | 8.5 |
| 2014 | Observed | 8.2 | 15.7 | 7.7 | 9.0 | 10.2 |
| | NORTRIP | 2.3 | 7.2 | 4.5 | 2.3 | 4.2 |
| | FORE | 9.2 | 11.2 | 5.3 | 3.7 | 8.0 |

Table 6. The selected numerical cases with maximal shares of light-duty vehicles using studded tyres and road maintenance measures for traction control considered by the two road dust emission models. The symbol + refers to 'included' and – to 'not included'.

| Model | Case | Abbreviation | Studded tyre share | Sanding | Salting |
|---|---|---|---|---|---|
| NORTRIP | 1 Reference | Ref | 80 % | + | + |
| | 2 Studded tyre share 50 % | ST 50 % | 50 % | + | + |
| | 3 Studded tyre share 30 % | ST 30 % | 30 % | + | + |
| | 4 Studded tyre share 0 % | ST 0 % | - | + | + |
| | 5 No sanding | no Sand | 80 % | - | + |
| | 6 No salting | no Salt | 80 % | + | - |
| FORE | 1 Reference | Ref | 80 % | + | - |
| | 2 Studded tyre share 50 % | ST 50 % | 50 % | + | - |
| | 3 Studded tyre share 30 % | ST 30 % | 30 % | + | - |
| | 4 Studded tyre share 0 % | ST 0 % | - | + | - |
| | 5 No sanding | no Sand | 80 % | - | - |

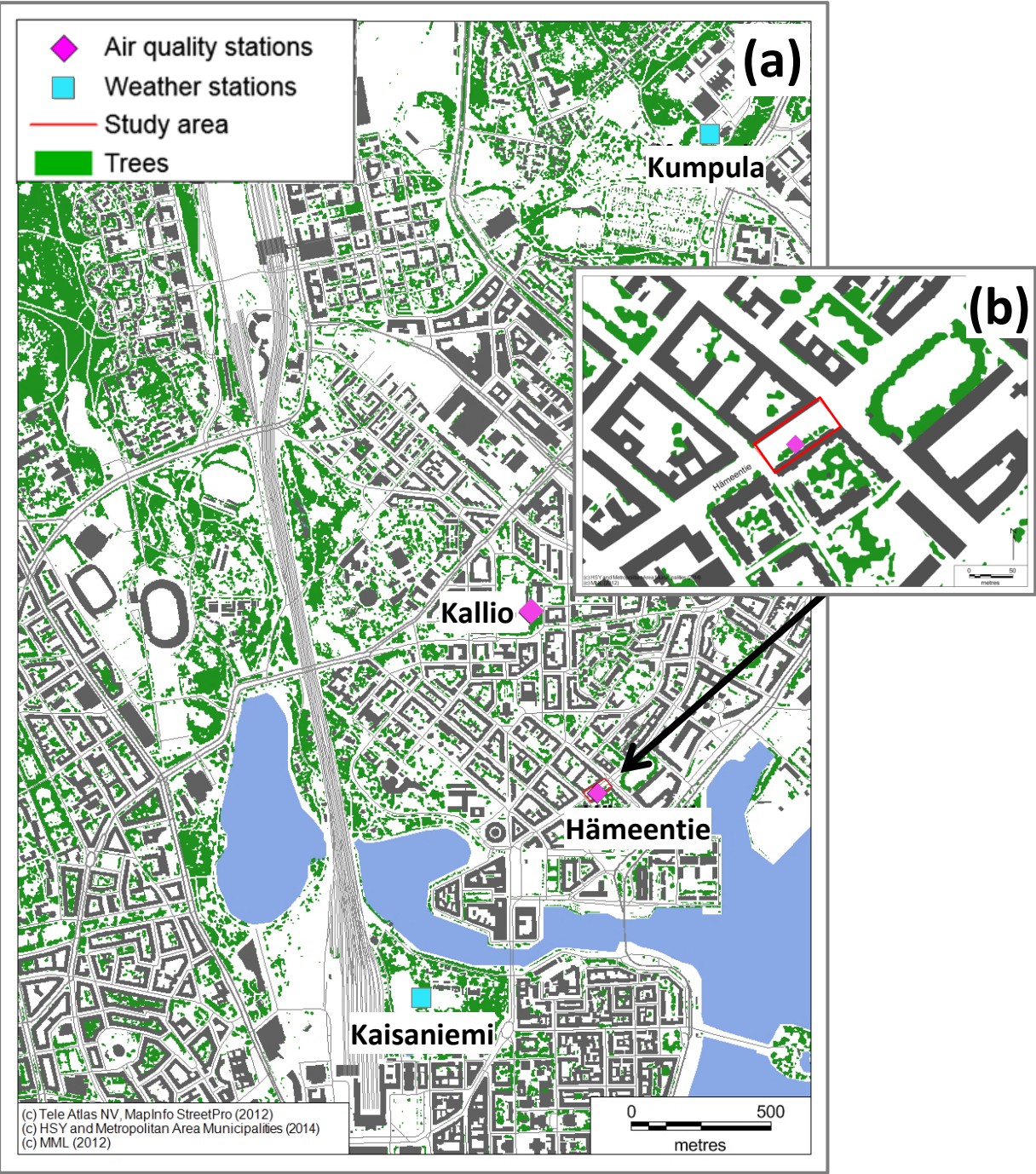

Figure 1. a) The locations of studied street segment (red rectangle) at Hämeentie, kerbside (Hämeentie) and urban background (Kallio) air quality stations (pink diamond), and weather stations (Kumpula and Kaisaniemi) (blue square) in central Helsinki. Trees have been marked with green circles. b) Close-up view showing building blocks (marked with grey colour) and trees in the vicinity of the studied street segment.

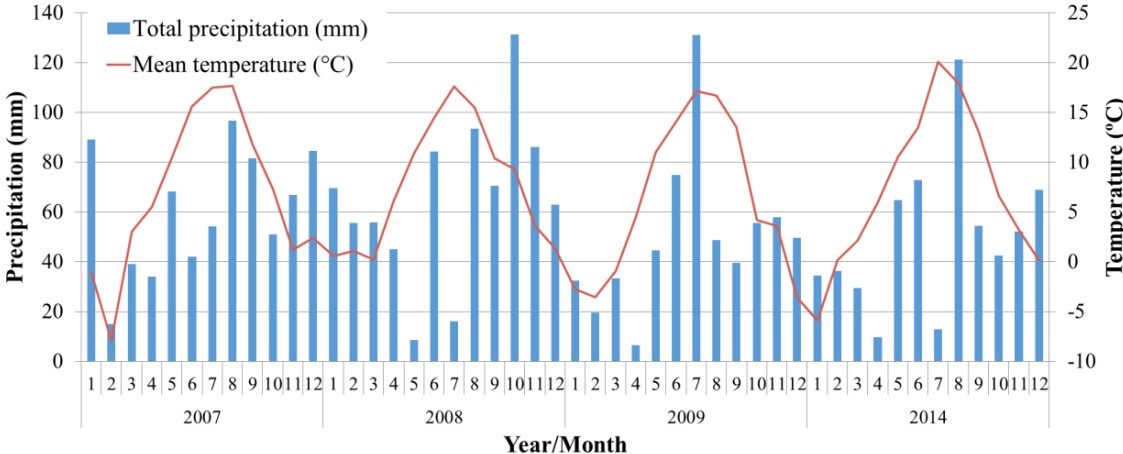

Figure 2. Monthly mean temperature (°C) and total precipitation (mm) for four years, measured at the meteorological station of Kaisaniemi.

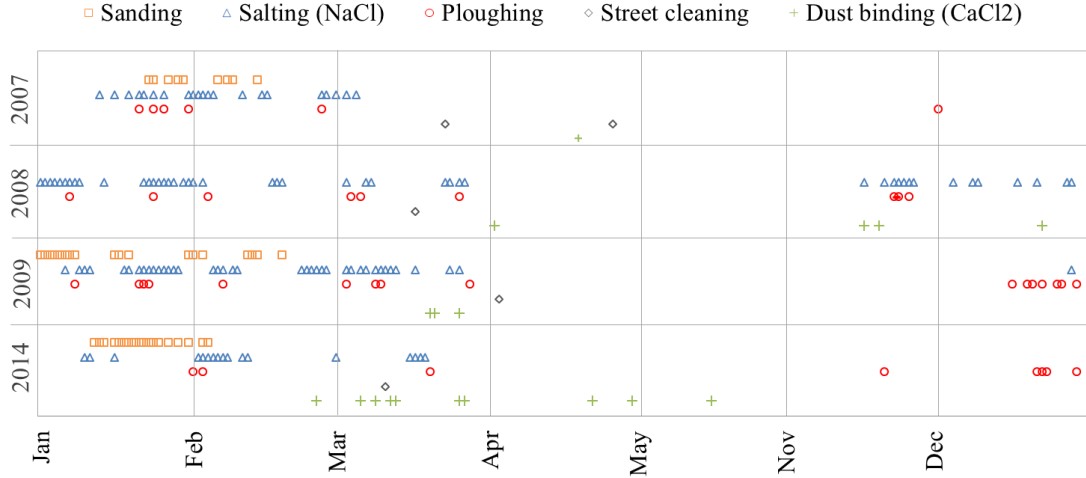

Figure 3. The approximate timing of the road maintenance measures at Hämeentie for four years. All the relevant information for the latter part of the year (from October to December) was available only for 2008.

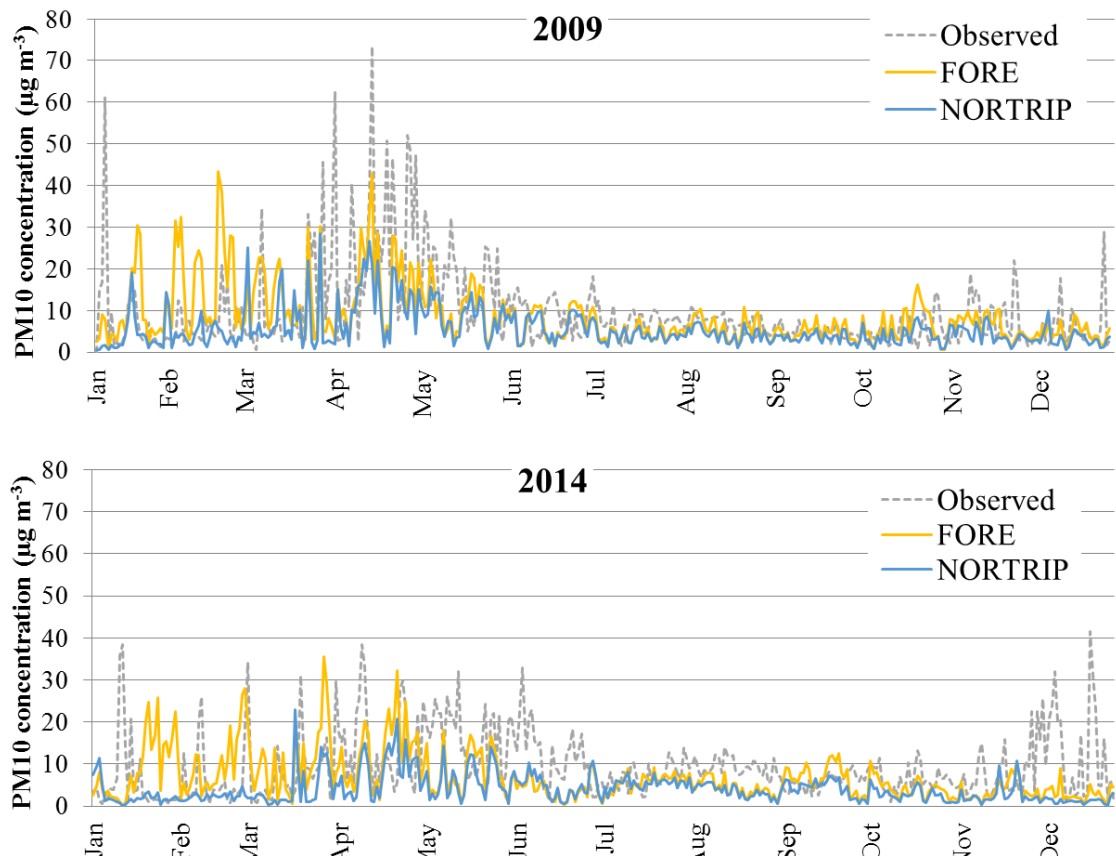

Figure 4. Time series of daily mean observed and modelled street increments of $PM_{10}$ for Hämeentie for 2009
(upper panel) and 2014 (lower panel).

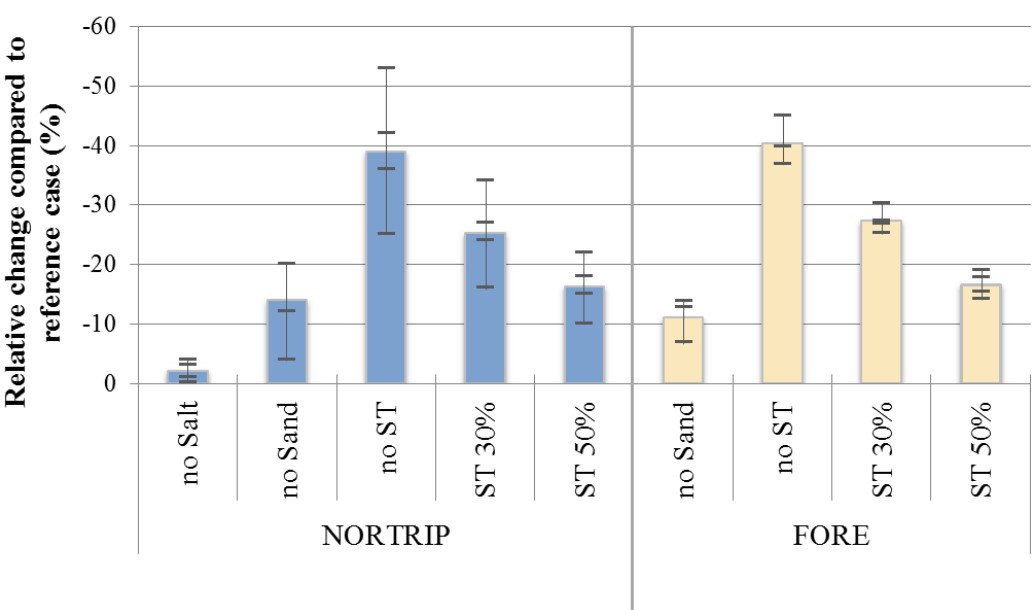

Figure 5.  Modelled relative changes in the non-exhaust street increment of $PM_{10}$ for the cases described in Table
6, averaged over four year period (2007-2009 and 2014). Line markers show values for the individual years.

