# Peer review of "The impact of measures to reduce ambient air $PM_{10}$"

_Atmospheric Chemistry and Physics, 2018_

## Referee Comment (RC1) · Anonymous Referee #1 · 13 Feb 2019

This paper has used road dust emission models to investigate the impact of studded tyre use on PM10 concentrations. The science is sound and the paper warrants publication once the following have been addressed:

Introduction In the Introduction it is mentioned that non-exhaust emissions are one of the most important causes of high roadside PM10 concentrations for several decades. However not details their overall contribution is given. Recent figures from the European Environment Agency state that ". In 2016, the non-exhaust emissions of PM2.5 constituted 42 % of emissions from the road transport sector, compared with 17 % in 2000 (for PM10, the contribution increased from 30 % in 2000 to 60 % in 2016)". https://www.eea.europa.eu/data-and-maps/indicators/transport-emissions-of-air-pollutants-8/transport-emissions-of-air-pollutants-6

Traffic data It's not clear what traffic speed was used in the models. A number of mentioned (weekday daytime, night-time, weekly and monthly). Given that emissions are speed dependent this is important. If there's increased braking and accelerating) this results in additional wear of both the tyre and the road surface. As such would one solution to reduce PM10 concentrations be a lower speed limit? It is noted that it is acknowledge that the NORTRIP model does not account for congested driving conditions but what likely error does this introduce?

Meteorological data How is snowfall taken into account with total precipitation?

Road maintenance data Are the roads washed during the summer? Street cleaning is shown in Table 3 but not Figure 3 Does snow ploughing have any impact?

The road dust emission model NORTRIP Need to justify why the amount of suspendable material in sand was set to 2%.

Evaluation of the vehicular exhaust emissions Given that the paper relates to PM10 emissions why not use PM10 emissions instead of those ofr PM2.5?

Results and discussion To save any confusion for readers specify seasons as winter (1 Jan to 14 March etc) Comparison of predicted and measured PM10 concentrations State the statistical significance of R2 values.

General discussion There should be some consideration of alternatives to road salt given the numerous papers which have highlighted the environmental impact of it. Studless winter tyres are becoming more popular – should Finland make this on option? There should be a discussion about the impact of different road surfaces on PM10 emissions (e.g. concrete, more durable asphalt). It is also important to highlight that the wear of the road surface increases with moisture level. Additionally after salting the road surface remains wet for longer periods and so road wear increases.

Typographical Check the spelling of "tyres" as in some places there is "tires". I would also prefer the use of "roads" rather than "pavements"

The road dust emission model FORE The model uses empirical reference emission factors which depend on the . . .. (note factors and depend)

---

## Referee Comment (RC2) · Anonymous Referee #2 · 15 Mar 2019

**General comments**

1.  This manuscript presents some interesting results, but the paper itself is currently written insufficiently well to be published in ACP. Examples of this include:
    a.  References to tables in the text without any explanation of what is in the table e.g. P3 L31 'The traffic data is summarized in Table 1'. A good quality paper would say what traffic data are summarised, and comment on the values in the table. *References to all tables and figures should be reviewed.*
    b.  The paper takes a standard format 'Intro, data, models etc'. However, information is not always provided in the right sections. e.g. some details of the modelling parameters are given in the introduction, and some more general text is given later in the paper, when it should come earlier. Some examples of where this has been done are given in the technical comments section below, but this list is not exhaustive. *The paper should be read carefully by the authors to make sure that all information is in the correct section.* This would make the paper easier to follow.
2.  There are some sections where insufficient information is provided regarding terminology. This makes comprehension difficult for a reader not familiar with the topic of Northern European non-exhaust. If terminology was better explained, the paper would be of interest to a wider readership.

**Specific comments**

3.  The title says road dust, but by P1 L17 the text talks about PM10 – and from then on the pollutant is also exclusively referred to as PM10. PM2.5 is mentioned later, but this distracts from the focus of the paper – if this is mentioned, more needs to be said on the proportion of PM10 that is PM2.5 during the episodes. There needs to be an explanation of how these relate; consider changing the manuscript title to refer to PM10.
4.  The measurements recorded at the study site need to be put into context early on in the paper e.g. values compared to EU and WHO AQ standards.
5.  The figures and tables should be improved to make the paper more attractive e.g.:
    a.  Figure 1 could be made less wide, so that there is an insert which shows more detail of the actual site – either in schematic form or a photograph. Increase text size. Hospitals are shown in pink not red.
    b.  Figure 2 is poor – consider text size and legend location.
6.  Include a summary table comparing the FORE and NORTRIP models, including the strengths and weaknesses. Mention dependence on parameters e.g. vehicle speed, traffic volume, HGV/LGV proportions.
7.  Section 3.2 on sensitivity analysis is poor – analyses that impact on both non-exhaust models need to be considered. e.g. different met inputs (precipitation), removing brake and tyre wear (this would only affect NORTRIP, but still is shows FORE is insensitive to this, and demonstrates the importance of this component).
8.  Consider adding more statistics or analyses that show the improved correlation of NORTRIP over FORE, e.g. an average diurnal profile?
9.  The section describing the NORTRIP model needs significant improvement (P5 L9-34):

a.  The paragraph needs to eb expanded to explain what has been taken from previous literature, how relevant these values are, and what assumptions are made in the model.
    b.  Is Boulter the right reference here - aren't these the EMEP factors?
    c.  How busy is the street? Can you comment on how accurate you think the brake and tyre wear factors are?
    d.  Sentence starting 'In model formulation...' is not a sentence.
    e.  L13 – say 'The road dust model emission calculation…'
    f.  L28 How does 'ploughing' relate to the activities in Table 2?
    g.  P29 provide reference for 2%.
10. P6 L15 This is a very high roughness value. Justify comparing to values in the literature for similar urban environments. Are you sure that this z0 represents the vicinity of the site, and is not just a value derived from the building heights on the street in question?
11. Section 2.1.3 Comment on the representativeness of the met for the study area. Could 'spring' be indicated on Figure 2; also the concentration time series should be put on this graph.

**Technical corrections**

12. P1 L18 says 'Both models', but 3 models have been listed.
13. Last two sentences of the abstract need to be made clearer.
14. P1 L31 and elsewhere, the manuscript refers to 'pavements' – usual English term is road.
15. Improve spelling e.g. tires in P1 L33.
16. Could mention street furniture (P1 L35)
17. P2 L12 say what temporal period winter tyres are mandatory.
18. P3 L7 – is 'it' building-to-building width?
19. P3L7 say why building height relevant, and provide approx. heights. State aspect ratio and comment on porosity.
20. P2 L8 say why the proportion of HDVs is relevant.
21. P2 L10-12 are the high trees really relevant? Doesn't the low roughness of the sea have more impact on dispersion that a few trees. If mentioning the trees can be justified, make sure they are clear in Fig. 1.
22. P2 L17-18 Information about what is done in the modelling should be in the modelling section.
23. P2 L31 The traffic data ARE … ditto P4 L4
24. P3 L35 which vehicles
25. P4 L1 Is this at the beginning or the end of the season?
26. P4 L6 usually refer to cloud cover rather than cloudiness
27. Section 2.1.4 1$^{st}$ para, should this be in the Introduction?
28. Section 3.1.4, last sentence – should be later in manuscript.
29. P4 L30 'made' not 'done'.
30. Section 2.1.5 re-word last sentence to make clearer.
31. P6 L4 justify use of reference emissions factors for the current study,
32. P6 L10/11 – sentence doesn't make sense.
33. P6 L12-14 explain further
34. P6 L16 Why quote/use different units if they are equivalent?

35. P6 L16 – 28 – Is any of this relevant to the NORTRIP model – if it is, it is in the wrong section.
36. P6 L16-19 – this section is supposed to be on FORE, not the study.
37. Section 2.2.2 Last sentence: explain why not and how much uncertainty this introduces e.g. is the traffic stop-start, or continuous?
38. General comment: mention if the traffic in the model given a time-varying emission profile. This may be important due to the non-linear relationship between emissions, meteorology and resultant concentrations.
39. P7 L3-4 How have these 9 street crossing geometries been taken into account?
40. P7 P6 – why mention NOx in this paper? Chemistry not relevant. Has this section of text been copied from elsewhere without consideration of its applicability to this particular application?
41. P7 L8-10 comment on meteorological and background pollutant data capture rates.
42. P7 L12 specify 'other' models
43. P7 L15-16 3 instances of poor punctuation and spelling.
44. P7 L19 – 'used as input to the model' rather than 'implemented'
45. P7 L21 say what the model sensitivity analyses demonstrate.
46. P7 L20-22 – refer to sections
47. P7 L28, first sentence – but this section starts with a time series?!
48. P7 L29 is this ACP notation?
49. Table 5 might be more interesting as a bar chart. Stats (NMSE, R, Fac2, Max values) could be of interest because the NORTRIP correlation is better than FORE.
50. P7 L38 – say what 'late' means for those less familiar with the cycle.
51. P8 L5 Improve sense.
52. P8 L11-14. These correlation stats are of interest. Are sub-daily observed concentration values available? Inspect the average diurnal variation to see if there is any relation to congestion.
53. P9 L1 May be non-linear due to congestion.
54. P9 L14-19 Doesn't need a table.
55. P9 L29 Compounds or configurations?
56. P9 last sentence – remove, not of relevance.
57. P10 L11 remind readers not in FORE.
58. P10 Don't need Table 8.
59. P10 around the 2$^{nd}$ paragraph – say something about the consistency of concentrations. If the models predict consistent concentrations why is there a massive difference in predictions in the Feb / March period?
60. P10 L19 which met parameters?
61. P10 L37 Us the term hypothetical?
62. P10 Can some emissions and concentration source apportionment analyses for the road be presented? i.e. in terms of road wear, tyre wear, brake wear, exhaust, resuspension, winter tyres – for both models?
63. Section 4, once other revisions to the paper have been completed , Section  should be reviewed e.g. P11 L14-15 – the substantial differences need to be made clearer.
64. P11 L17 Yes!

65. Table 3 – where does the '5 and 10' referred to in the caption come from? What does the last sentence in the caption mean? How do these values relate to the road? Are any of these values from EMEP?
66. Table 5 (possibly elsewhere) as the models are introduced as NORTRIP then FORE, they should be presented accordingly in the table.
67. Figure 4 – explain why FORE does badly Jan-Mar i.e. predicts much larger values than NORTRIP for that period.
68. Reduce the caption length for Figure 5.

---

## Author Comment (AC1) · 10 May 2019

**Response to anonymous Reviewer #1**

We would like to thank the anonymous Reviewer #1 for his/her comments and suggestions for improving this manuscript. Our response to the reviewer's comments is provided below. The reviewer's comments are written in italic.

*This paper has used road dust emission models to investigate the impact of studded tyre use on PM10 concentrations. The science is sound and the paper warrants publication once the following have been addressed:*

*Introduction In the Introduction it is mentioned that non-exhaust emissions are one of the most important causes of high roadside PM10 concentrations for several decades. However not details their overall contribution is given. Recent figures from the European Environment Agency state that ". In 2016, the non-exhaust emissions of*
*PM2.5 constituted 42 % of emissions from the road transport sector, compared with17 % in 2000 (for PM10, the contribution increased from 30 % in 2000 to 60 % in2016)". https://www.eea.europa.eu/data-and-maps/indicators/transport-emissions-ofair-pollutants-8/transport-emissions-of-air-pollutants-6*
**Answer:**
The reviewer is correct. Overall contribution from the suggested source has been included in the revised manuscript with a sentence: 'Estimated relative contribution of non-exhaust emissions to the emissions of $PM_{10}$ from road transport increased from 30 % in 2000 to 60 % in 2016.' Reference to the European Environment Agency will be included.

*Traffic data It's not clear what traffic speed was used in the models. A number of mentioned (weekday daytime, night-time, weekly and monthly). Given that emissions are speed dependent this is important. If there's increased braking and accelerating) this results in additional wear of both the tyre and the road surface. As such would one solution to reduce PM10 concentrations be a lower speed limit? It is noted that it is acknowledge that the NORTRIP model does not account for congested driving conditions but what likely error does this introduce?*
**Answer:**
In order to clarify derivation of the traffic speed for Hämeentie, we have rewritten the sentence in question in the following manner: 'The vehicle speeds for the night-time hours and weekend days were evaluated using the measured diurnal and weekly cycles of vehicle speeds in Runeberginkatu (located 2 km southwest from Hämeentie) in 2004.' Fig. 1 below shows derived average diurnal cycle of traffic volume and speed for years 2007-2009 and 2014. Modelled emissions and PM10 concentrations resemble the diurnal cycle of traffic volume.

We agree with the reviewer that the reduction of the speed limit would be a potential abatement measure for the ambient air PM10 concentrations. However, for Hämeentie where average daily vehicle speed is already very low (26 km/h), and its further reduction would not be possible in practise. We therefore considered it more important to investigating impact of studded tyre reduction and traction control measures.

The congestion could be a source of error but no measurements are available to quantify this. The treatment of the road wear in the NORTRIP model is based on vehicular speed and not on acceleration. Additionally, as acknowledged in the section 3.2.1 of the revised manuscript, the form of the dependency of road wear on vehicle speed in low speed conditions is uncertain.

[Figure]

Figure 1. Diurnal cycle of traffic volume and vehicle speed at Hämeentie averaged over the four years (2007-2009 and 2014).

*Meteorological data How is snowfall taken into account with total precipitation?*
**Answer:**
In NORTRIP model input, precipitation is included as either rain (mm/h) or snow (mm/h). In case when only information about the total precipitation is available, snowfall is defined as being precipitation for atmospheric temperatures below zero. However, if data is available separately for the precipitation of rain and snow, these values can be used as such in the NORTRIP model input. The FORE model allows one input value for precipitation (mm/h), i.e., it does not separate between solid and liquid water precipitation. We will include a description of these to the revised manuscript.

*Road maintenance data Are the roads washed during the summer? Street cleaning is shown in Table 3 but not Figure 3 Does snow ploughing have any impact?*
**Answer:**
Information about the street cleaning and ploughing events will be added to the Figure 3 in the revised manuscript. Street cleaning is typically conducted after the cold season, as soon as weather permits, i.e. when freezing temperatures subside in spring. For the years considered in this study, all street cleaning activities took place from mid-March until the end of April. In the road dust emission modelling, ploughing was taken into account only by the NORTRIP model. Ploughing reduces the amount of dust on the street surface with the predefined efficiency factor, which is expected to be very low. In the NORTRIP model application for Hämeentie, this is set to be 1% and 0.1% for the non-suspendable and suspendable fraction of the road dust, respectively. The same efficiency has been assumed for the street cleaning. We have included a description of the street cleaning and ploughing efficiencies to the section 2.2.1 in the revised manuscript.

*The road dust emission model NORTRIP Need to justify why the amount of suspendable material in sand was set to 2%.*
**Answer:**
Measurements of the sand size distribution are required in order to identify fraction of the sanding material that is available for suspension. The data concerning this aspect of sanding material quality is often limited, if available at all. In this study we have used information about the size distribution for sand used in Helsinki Metropolitan Area reported in Kulovuori et al. (2019) that have found amount of suspendable fraction (<200µm) to range from 0.4 % to 2.5 %. Lower suspendable fraction has been found for wet sieved sanding material and higher for the sand with unknown sieving status. We have assumed value of 2%. The reference has been added to the revised manuscript and quality of sanding material will been discussed in Section 3.3.

*Evaluation of the vehicular exhaust emissions given that the paper relates to PM10 emissions why not use PM10 emissions instead of those of PM2.5?*
**Answer:**
LIPASTO emission modelling system that was used for evaluation of the exhaust emissions does not separate $PM_{10}$ and $PM_{2.5}$ exhaust particle emissions. These emissions are addressed as 'exhaust particulate matter emissions', i.e. 'exhaust PM emissions'. Only a very small fraction (or none) of the exhaust emissions are in the coarse particle range (larger than 2.5 micrometres). This has been corrected in the revised manuscript.

*Results and discussion To save any confusion for readers specify seasons as winter (1 Jan to 14 March etc)*
**Answer:**
The suggested correction has been included in the revised manuscript.

*Comparison of predicted and measured PM10 concentrations state the statistical significance of R2 values.*
**Answer:**
More detailed statistical analyses, and their interpretation, have been included as annex in the revised manuscript.

*General discussion There should be some consideration of alternatives to road salt given the numerous papers which have highlighted the environmental impact of it.*
**Answer:**
In Finland, sanding is considered as the main alternative traction control method in the areas with sensitive environment to the use of salt (e.g., in areas, in which the ground water supplies could be contaminated). The use of wood chips has been examined by the Finnish Transport Infrastructure Agency as an alternative traction control method but only for the bicycle lanes. This will be mentioned in the section 3.3 in the revised manuscript.

Studless winter tyres are becoming more popular – should Finland make this on option?
**Answer:**
The reduction of studded tyre use is a feasible option for the road dust abatement, also in Finland. However, policy measures, such as studded tyre charges in Norway or studded tyre ban in individual streets in Sweden, have not been introduced in Finland. Non-studded winter tyres have not gained a wider popularity among drivers, apart from their moderate increasing trend in the Helsinki metropolitan area. Average share of light duty vehicles with studded tyres observed between December and February (inclusive) decreased from 80% in the season2014/2015 to 75% in the season 2018/2019 in favour of the non-studded winter tyres.

In this study, we have demonstrated the potential to reduce the impact of non-exhaust traffic induced particle emissions on ambient air PM10 concentrations, with transition from studded to non-studded winter tyres (Section 3.3). In studied cases with reduced number of vehicles using studded tyres, studded tyres were reduced in favour of non-studded winter tyres. More discussion will be added on this subject to the revised manuscript.

*There should be a discussion about the impact of different road surfaces onPM10 emissions (e.g. concrete, more durable asphalt). It is also important to highlight that the wear of the road surface increases with moisture level. Additionally after salting the road surface remains wet for longer periods and so road wear increases.*
**Answer:**
The reviewer is correct. We have expanded Section 3.2.1 with more information about the wear rates used in this study in context of the road surface characteristics. At the same time, impact of road surface moisture on the wear is mentioned. Added paragraph is as follows:' In this study, we have used wear rates derived for the reference pavement type (SMA16 with porphyry from Älvdalen) in the Swedish road wear model (Jacobson and Wågberg, 2007) which is one of the most wear resistant asphalt pavements used in Sweden. The wear rates in the Swedish road wear model are based on laboratory and field experiments and provide an average under both dry and wet conditions. However, influence of surface moisture that increases the wear is not

directly considered in the model calculations. Denby et al. 2013a estimated the typical wear rates to be from 2 to 5 g km$^{-1}$ veh$^{-1}$ and acknowledged significantly variation of these values depending on the material used with increased wear rates for roads with the poor quality surfaces.'

Additionally, improvement of pavement quality in terms of the rock aggregate size and durability, or use of alternative pavements has been mentioned as a factor that will enhance air quality benefits along with the studded tyre reduction. Following paragraph has been added to the section 3.3: 'The effect of the studded tyres reductions can be enhanced by improving the quality of road surfaces. Larger aggregate size from rocks that are more resistant to wear in the asphalt pavements, or use of alternative pavements can reduce PM10 emissions (Gustafsson et al. 2009; Gustafsson and Johansson 2012)  and therefore, have positive effect on the ambient air PM10 concentrations.'

*Typographical Check the spelling of "tyres" as in some places there is "tires". I would also prefer the use of "roads" rather than "pavements"*

**Answer:**

Suggested corrections regarding the spelling and terminology have been included in the revised manuscript, e.g. 'pavement wear' has been replaced with 'road wear'.

*The road dust emission model FORE The model uses empirical reference emission factors which depend on the : : :. (note factors and depend)*

**Answer:**

The sentence in question has been replaced with the following: 'The model uses empirical reference emission factors, which have different values depending on the time of the year, the mass fraction of particles (PM$_{10}$ or PM$_{2.5}$), and the traffic environment (urban or highway). The reference emission factor will be higher for the time of the year when sanding and studded tyres are commonly used (referred to as 'sanding period') compared to the rest of the year ('non-sanding period').'

References:

Gustafsson, M., Blomqvist, G., Gudmundsson, A., Dahl, A., Jonsson, P., Swietlicki, E.: Factors influencing PM10 emissions from road pavement wear. Atmospheric Environment 43 (31), 4699e4702, 2009.

Gustafsson, M. and Johansson, C.: Road pavements and PM10. Summary of the results of research funded by the Swedish Transport Administration on how the properties of road pavements influence emissions and the properties of wear particles, Trafikverket, Report 2012:241, 2012.

Kulovuori, S., Ritola, R., Stojiljkovic, A., Kupiainen, K., Malinen, A.: Katupölyn lähteet, päästövähennyskeinot ja ilmanlaatuvaikutukset – Tuloksia KALPA-tutkimushankkeesta 2015–2018. HSY publications 1/2019, in Finnish, summary in Endglish, URL: https://www.hsy.fi/sites/Esitteet/EsitteetKatalogi/Julkaisusarja/1-2019-katupolyn-lahteet-paastovahennyskeinot-ja-ilmanlaatuvaikutukset-KALPA-2015-2018.pdf

---

## Author Comment (AC2) · 10 May 2019

**Response to anonymous Reviewer #2**

We would like to thank the anonymous Reviewer #2 for his/her comments and suggestions for improving this manuscript. Our response to the reviewer's comments is provided below. The reviewer's comments are written in italic.

*General comments*
*1. This manuscript presents some interesting results, but the paper itself is currently written insufficiently well to be published in ACP. Examples of this include:*
*a. References to tables in the text without any explanation of what is in the table e.g. P3 L31 'The traffic data is summarized in Table 1'. A good quality paper would say what traffic data are summarised, and comment on the values in the table. References to all tables and figures should be reviewed.*
**Answer:**

The reviewer is correct. We have carefully checked the presentation of all the figures and tables, and added a proper discussion to all of these.

We have also removed one table that contained results that were already presented in a figure (Table 8 in the reviewed version of the manuscript). The key information in Table 6 in the reviewed manuscript was condensed to one sentence of text, and that table was removed.

*b. The paper takes a standard format 'Intro, data, models etc'. However, information is not always provided in the right sections. e.g. some details of the modelling parameters are given in the introduction, and some more general text is given later in the paper, when it should come earlier. Some examples of where this has been done are given in the technical comments section below, but this list is not exhaustive. The paper should be read carefully by the authors to make sure that all information is in the correct section. This would make the paper easier to follow.*
**Answer:**

We have carefully checked the whole manuscript, and tried our best to place all the information into the proper sections.

For instance, the comments on when winter tyres are obligatory were removed from the introduction (this has been presented in section 2.1.2), and part of the text that was in 'methods' (containing results) has been moved to 'results'.

*2. There are some sections where insufficient information is provided regarding terminology. This makes comprehension difficult for a reader not familiar with the topic of Northern European non-exhaust. If terminology was better explained, the paper would be of interest to a wider readership.*
**Answer:**

Regarding 'the street increment of $PM_{10}$', we have added a brief characterisation to the abstract. The detailed definition of this concept is in the beginning of section 3 (Results). Additionally, explanation of the non-exhaust increment of $PM_{10}$ has been added to section 3 in the revised manuscript.
However, it is not totally clear to us, which concepts the reviewer is referring to; key concepts of non-exhaust emissions are given in the introduction.

*Specific comments*
*3. The title says road dust, but by P1 L17 the text talks about PM10 – and from then on the pollutant is also exclusively referred to as PM10. PM2.5 is mentioned later, but this distracts from the focus of the paper – if this is mentioned, more needs to be said on the proportion of PM10 that is PM2.5 during the episodes. There needs to be an explanation of how these relate; consider changing the manuscript title to refer to PM10.*
**Answer:**

We have improved the title of the manuscript (as per reviewer comment) to be more specific, the revised version reads: "… ambient air $PM_{10}$ concentrations originated from road dust …. ". In the whole of the manuscript, we have also checked the terminology throughout the text. It is made clear in the revised version that we address $PM_{10}$. All the references to $PM_{2.5}$ are removed as unnecessary.

*4. The measurements recorded at the study site need to be put into context early on in the paper e.g. values compared to EU and WHO AQ standards.*

**Answer:**

In this study, we have focused on the street increment $PM_{10}$ concentrations; thus we have not addressed measured kerbside $PM_{10}$ concentrations in the context of the EU and WHO air quality standards. However, paragraph has been added to the section 3.1 in the revised manuscript relating measurements to the air quality standards, and total observed kerbside $PM_{10}$ concentrations will be included in Table 5 together with the street increment concentrations.

*5. The figures and tables should be improved to make the paper more attractive e.g.:*
*a. Figure 1 could be made less wide, so that there is an insert which shows more detail of the actual site – either in schematic form or a photograph. Increase text size. Hospitals are shown in pink not red.*
*b. Figure 2 is poor – consider text size and legend location.*

**Answer:**

We have carefully re-drawn all the figures in the manuscript. We have also added a new panel to Fig. 1 that shows the immediate surroundings of the measurement site. In our view, all the revised figures are more attractive and understandable, and also technically of a better quality.

*6. Include a summary table comparing the FORE and NORTRIP models, including the strengths and weaknesses. Mention dependence on parameters e.g. vehicle speed, traffic volume, HGV/LGV proportions.*

**Answer:**

We have included summarised comparison of the two road dust emission models in the revised manuscript, however, in form of a paragraph. We have chosen this approach since it would be difficult to directly compare these two models that are very much conceptually different, i.e. use different approaches in the calculation of the non-exhaust emissions. A detailed inter-comparison of the structures of the two models would be an extensive task, and that is outside the scope of this study. The included paragraph is as follows: 'The NORTRIP model is a process based model that describes wear processes, traffic induced suspension of accumulated road dust and impact of road maintenance activities (salting, sanding, dust binding, cleaning and ploughing) on both dust load and road surface moisture. It includes dependences on vehicle speed, tyre type, vehicle category (light and heavy duty vehicles) and road surface properties that enable comprehensive evaluation of the road dust abatement measures. The NORTRIP model requires extensive input data that is often not available (e.g. road maintenance data, properties of the road surface). This may present a challenge in application of the NORTRIP model. The FORE model requires relatively much less input data. However, it relies on the reference emission factors which need to be computed based on the local air quality measurements. The FORE model considers two road dust sources, viz. road wear and traction sand. The model does not account for the dependence of emissions on vehicle speed and traffic fleet composition, which limits application of the model for the evaluation of a wider range of measures to reduce road dust.'

*7. Section 3.2 on sensitivity analysis is poor – analyses that impact on both non-exhaust models need to be considered. e.g. different met inputs (precipitation), removing brake and tyre wear (this would only affect NORTRIP, but still is shows FORE is insensitive to this, and demonstrates the importance of this component).*

**Answer:**

It is correct that this section was actually not at all a (comprehensive) sensitivity analysis. We have re-written the whole section, and also replaced the previous title by a more descriptive one, i.e., "Evaluation of the uncertainties of the modelling". Our aim here was to analyse both qualitatively and in part quantitatively how large the effects of various key sources of uncertainty could be, but not to present a comprehensive and harmonised sensitivity analysis.

The main aim of this study was to evaluate the effectiveness of mitigation options, not to present a thorough sensitivity analysis. Such an analysis, in a harmonised manner for the both suspension emission models, and for the OSPM model, would be a very extensive research task. For instance, a sensitivity analyses solely for the OSPM model have already been previously published. Sensitivity of the NORTRIP model to a range of

parameters (including meteorological parameters) has been studied by Denby et al. 2013. Sensitivity of the FORE model to precipitation and traction sanding input is available from by Kauhaniemi et al. 2011.

*8. Consider adding more statistics or analyses that show the improved correlation of NORTRIP over FORE, e.g. an average diurnal profile?*

**Answer:**

The R2 values of hourly mean concentrations have been added in the revised manuscript.
More detailed statistical analyses, and their interpretation, have been included as annex in the revised manuscript.

*9. The section describing the NORTRIP model needs significant improvement (P5 L9-34):*
*a. The paragraph needs to eb expanded to explain what has been taken from previous literature, how relevant these values are, and what assumptions are made in the model.*

**Answer:**

We have revised the section describing the NORTRIP model and tried our best to improve it according to the reviewer's suggestion, however keeping the description brief. The reader is therefore referred to other sources for more detailed description of the model. For example, we clarified that used brake and tyre wear rates were taken from the literature; we included more information about the derivation of the suspension factor; and we added assumption made for the efficiency of ploughing and street cleaning for the removal of the road dust.

*b. Is Boulter the right reference here - aren't these the EMEP factors?*

**Answer:**

Boulter is the correct reference that has also been used in the NORTRIP model documentation.

*c. How busy is the street? Can you comment on how accurate you think the brake and tyre wear factors are?*

**Answer:**

Tyre and brake weak factors were taken from Boulter (2005) which provides a review of tyre and brake emission factors, generally, with a significantly wide range of values. In the NORTRIP model reasonable values have been taken within this range. It would be possible to, for example, investigate accuracy of modelled contribution of the brake and tyre wear against the source apportionment studies for the ambient air $PM_{10}$ samples, e.g., using the method described by Kupiainen et al. 2016. Unfortunately, such analyses were not available.

*d. Sentence starting 'In model formulation...' is not a sentence.*

**Answer:**

The sentence in question has been revised in the following manner: 'The road wear and suspension are assumed to be linearly dependent on vehicle speed.'

*e. L13 – say 'The road dust model emission calculation…'*

**Answer:**

The suggested correction has been included in the revised manuscript.

*f. L28 How does 'ploughing' relate to the activities in Table 2?*

**Answer:**

Information about ploughing events during the study period has been included in the Table 2 and in the Figure 3 in the revised manuscript.

*g. P29 provide reference for 2%.*

**Answer:**

Measurements of the sand size distribution are required in order to identify fraction of the sanding material that is available for suspension. The data concerning this aspect of sanding material quality is often limited, if available at all. In this study we have used information about the size distribution for sand used in Helsinki Metropolitan Area reported in Kulovuori et al. (2019) that have found amount of suspendable fraction (<200µm) to range from 0.4 % to 2.5 %. Lower suspendable fraction has been found for wet sieved sanding material and higher for the sand with unknown sieving status. We have assumed value of 2%. The reference has been added to the revised manuscript and quality of sanding material will been discussed in Section 3.3.

*10. P6 L15 This is a very high roughness value. Justify comparing to values in the literature for similar urban environments. Are you sure that this z0 represents the vicinity of the site, and is not just a value derived from the building heights on the street in question?*

**Answer:**

The average building height is 26 m on the studied street canyon (including study site and immediate surroundings). Thus, the value of z0 is derived from the average building height in the street canyon in question. The value of z0 represents local conditions (i.e. studied street canyon and immediate neighbourhood within the distance scale of about hundred meters), not wider area (e.g. scale of kilometres) around the studied street canyon.

*11. Section 2.1.3 Comment on the representativeness of the met for the study area. Could 'spring' be indicated on Figure 2; also the concentration time series should be put on this graph.*

**Answer:**

Comment regarding the representativeness of the meteorological data has been included in the revised manuscript with the paragraph: 'Using meteorological data at two urban stations could result in reduced representatives of the micrometeorological processes. Particularly, small-scale rain showers could be detected at the urban meteorological stations, but not at the study site, or the other way around.'

The observed street increment of $PM_{10}$ has been presented in Fig. 4 for years 2009 and 2014. We felt that as Fig. 2 is in our view busy as it is (already a curve and a histogram, with two vertical axes), adding additional concentration and other information could make it more difficult to read. The total observed $PM_{10}$ concentrations will be included in table 5 as answer to the reviewer's comment #4.

***Technical corrections***
*12. P1 L18 says 'Both models', but 3 models have been listed.*

**Answer:**

In order to clarify to which models we imply, sentence has been modified in the revised manuscript.

*13. Last two sentences of the abstract need to be made clearer.*

**Answer:**

Last two sentences of the abstract have been modified in the revised manuscript as follows: 'Modelled contributions of traction sand and salt to the mean annual non-exhaust increment of $PM_{10}$ ranged from 4% to 20% for the traction sand, and from 0.1 % to 4 % for the traction salt. The results presented here can be used to support development of optimal strategies for reducing the high springtime particulate matter concentrations originated from the road dust.'

*14. P1 L31 and elsewhere, the manuscript refers to 'pavements' – usual English term is road.*

**Answer:**

The suggested correction has been included in the revised manuscript.

*15. Improve spelling e.g. tires in P1 L33.*

**Answer:**

The corrections in spelling have been included in the revised manuscript.

*16. Could mention street furniture (P1 L35)*

**Answer:**

In the sentence in question which describes how the road dust particles are commonly formed, street furniture can be considered to be part of the surrounding environment that was listed as a more general miscellaneous source.

*17. P2 L12 say what temporal period winter tyres are mandatory.*

**Answer:**

The sentence in question has been removed in the revised manuscript. However, temporal period when winter tyres are mandatory is included in section 2.1.2.

*18. P3 L7 – is 'it' building-to-building width?*

**Answer:**

Building-to-building width of the selected segment of Hämeentie is 32 m. Description of the site has been improved in the revised manuscript.

*19. P3L7 say why building height relevant, and provide approx. heights. State aspect ratio and comment on porosity.*

**Answer:**

Information about the aspect ratio and porosity of the street canyon has been included in the revised manuscript in the Section 2.2.3.

*20. P3 L8 say why the proportion of HDVs is relevant.*

**Answer:**

We have placed information about the HDVs share to section 2.1.2. Relevance of the HDVs will be mentioned in the revised manuscript.

*21. P3 L10-12 are the high trees really relevant? Doesn't the low roughness of the sea have more impact on dispersion that a few trees. If mentioning the trees can be justified, make sure they are clear in Fig. 1.*

**Answer:**

Section 2.1.1 describes the structure of the studied street canyon (i.e. building heights and in this case also tree heights) which is used for construction of the street canyon in OSPM model. Thus, in this case, the tree heights are relevant, to properly construct the street canyon right next to the studied street segment. It is not possible to provide roughness value for OSPM and the model considers only the street canyon in question, but not the surroundings of the street. Thus, the sea area that is located several hundred meters from the canyon cannot be considered in model input. Fig. 1 has been revised to include trees.

*22. P3 L17-18 Information about what is done in the modelling should be in the modelling section.*

**Answer:**

The sentence in question has been modified in the revised manuscript as follows: 'The average height of the surrounding buildings in the studied segment of the street is 26 m.'

*23. P3 L31 The traffic data ARE … ditto P4 L4*

**Answer:**

The suggested correction has been included in the revised manuscript.

*24. P3 L35 which vehicles*

**Answer:**

Studded tyres are used only on the light duty vehicles. This has been clarified in the revised manuscript.

*25. P4 L1 Is this at the beginning or the end of the season?*

**Answer:**

The transition between summer and winter tyres happens at the beginning and at the end of the winter tyre season. In order to clarify this in the revised manuscript, the sentence in question has been replaced with the following: 'For other considered years (2007-2009), such detailed information was not available, and the winter tyre season was therefore set to last from 23 October until 30 April. The transition between winter and summer tyres is assumed to be linear over a one-month period at the beginning and at the end of the winter tyre season.'

*26. P4 L6 usually refer to cloud cover rather than cloudiness*

**Answer:**

The suggested correction has been included in the revised manuscript.

*27. Section 2.1.4 1st para, should this be in the Introduction?*

**Answer:**

The paragraph has been moved to the Introduction in the revised manuscript.

*28. Section 2.1.4, last sentence – should be later in manuscript.*

**Answer:**

The last to sentences have been removed in the revised manuscript. Information about the road maintenance considered by the road dust emission models has been included in the section 2.2.1.

*29. P4 L30 'made' not 'done'.*

**Answer:**

The suggested correction has been included in the revised manuscript.

*30. Section 2.1.5 re-word last sentence to make clearer.*

**Answer:**

The sentence in question has been moved to the results section in the revised manuscript. The sentence has been rewritten as follows: 'The urban background contribution to the concentrations measured at the street level in Hämeentie was substantial, i.e., 64%, averaged over the two years with available data (2009 and 2014).'

*31. P6 L4 justify use of reference emissions factors for the current study,*

**Answer:**

The paragraph regarding the reference emission factor in FORE has been revised and currant version reads: 'We have adopted the reference emission factors evaluated for Stockholm estimated and further explained by Omstedt et al. (2005); i.e., 1200 and 200 µg veh$^{-1}$ m$^{-1}$, for the sanding (Oct-May) and non-sanding (Jun-Sep) period, respectively. The climatic conditions, studded tyre shares and the procedures of using traction sand are fairly similar in Stockholm and Helsinki, although the difference in the used amounts of sand and salt can be significant.'

*32. P6 L10/11 – sentence doesn't make sense.*

**Answer:**

The sentence in question has been modified in the revised manuscript as follows: 'The dust layer is reduced by the resuspension of particles due to the air flow and by runoff due to precipitation.'

*33. P6 L12-14 explain further*

**Answer:**

The sentences in question have been replaced with the following:

'The suspension of road dust particles in the air is controlled by road surface moisture that is based on modelling of precipitation, runoff, and evaporation. The effect of terrain on wind is defined by roughness length which is needed for the evaluation of the evaporation (Omstedt et al. 2005).'

*34. P6 L16 Why quote/use different units if they are equivalent?*
**Answer:**

The units have been removed in the revised manuscript.

*35. P6 L16 – 28 – Is any of this relevant to the NORTRIP model – if it is, it is in the wrong section.*
**Answer:**

Adjustment of the studded tyre share has been done only for FORE model. The content has been replaced in the revised manuscript with the following paragraph: 'The model does not allow for the dependencies of emissions on vehicle speed and fleet composition. As studded tyres are only used in light duty vehicles (LDV's), the share of studded tyres in the total traffic fleet is relatively lower in Hämeentie. In the FORE model, we have used as input the studded tyre share of the whole traffic fleet of Hämeentie, including both light duty and heavy duty vehicles. For instance, assuming that 80%, 50%, 30% or 0% of the LDV's uses studded tyres, the studded tyre share of the whole traffic fleet is approximately 57%, 35%, 21% and 0%, respectively.'

*36. P6 L16-19 – this section is supposed to be on FORE, not the study.*
**Answer:**

The reviewer is correct. Sentences describing the study site have been removed in the revised manuscript.

*37. Section 2.2.2 Last sentence: explain why not and how much uncertainty this introduces e.g. is the traffic stop-start, or continuous?*

**Answer:**

Detailed information about the speed dependence of PM exhaust emission factors was not available in LIPASTO data base. The LIPASTO exhaust emission factors were only available for so-called "street" and "road" speeds. These so-called "street speed" emission factors consider non-continuous driving style.

*38. General comment: mention if the traffic in the model given a time-varying emission profile. This may be important due to the non-linear relationship between emissions, meteorology and resultant concentrations.*

**Answer:**

Yes, the modelled non-exhaust and exhaust emissions have time-varying profile, due to the dependency on the hourly traffic data. This has been mentioned in the revised manuscript.

*39. P7 L3-4 How have these 9 street crossing geometries been taken into account?*

**Answer:**

The geometry of street crossings is considered by wind sectors and so-called building height exceptions in the OSPM model.

*40. P7 P6 – why mention NOx in this paper? Chemistry not relevant. Has this section of text been copied from elsewhere without consideration of its applicability to this particular application?*

**Answer:**

The sentence has been removed in the revised manuscript.

*41. P7 L8-10 comment on meteorological and background pollutant data capture rates.*

**Answer:**

The comment on meteorological and background pollutant data capture rates has been included with two sentences: 'Meteorological and background concentration data used for the OSPM calculations were very well covered. Data coverage for wind speed and direction, and background concentrations was 97% and 98%, respectively.'

*42. P7 L12 specify 'other' models*

**Answer:**

We have revised section 2.2.3 according to the reviewer's comments 19, 21 and 29-41. Significant modifications have been made and sentence in question has been removed in the revised manuscript.

*43. P7 L15-16 3 instances of poor punctuation and spelling.*

**Answer:**

We have revised section 2.2.3 according to the reviewer's comments 19, 21 and 29-41. Significant modifications have been made and sentence in question has been removed in the revised manuscript.

*44. P7 L19 – 'used as input to the model' rather than 'implemented'*

**Answer:**

The suggested correction has been included in the revised manuscript.

*45. P7 L21 say what the model sensitivity analyses demonstrate.*

**Answer:**

As answered to reviewer's comment #7, we have re-written the whole section, and also replaced the previous title by a more descriptive one, i.e., "Evaluation of the uncertainties of the modelling". We have analysed and numerically evaluated selected key uncertainties related to the application of the two road dust emission models, and to the street canyon modelling for the Hämeentie site. This has been stated in the section 3.2.

*46. P7 L20-22 – refer to sections*

**Answer:**

References to sections have been added in the revised manuscript.

*47. P7 L28, first sentence – but this section starts with a time series?!*

**Answer:**

The notation mentioned in the reviewer's comment #48 has been corrected.

*48. P7 L29 is this ACP notation?*

**Answer:**

The notation has been corrected and the sentence in the revised manuscripts reads as follows: 'Seasons were defined as follows: winter (1 Jan to 14 Mar), spring (15 Mar to 31 May), summer (1 Jun to 30 Sep) and autumn (1 Oct to 31 Dec).

*49. Table 5 might be more interesting as a bar chart. Stats (NMSE, R, Fac2, Max values) could be of interest because the NORTRIP correlation is better than FORE.*

**Answer:**

More detailed statistical analyses, and their interpretation, have been included as annex in the revised manuscript. However, we will keep the mean annual and seasonal observed and modelled street increments of $PM_{10}$ as a table.

*50. P7 L38 – say what 'late' means for those less familiar with the cycle.*

**Answer:**

The sentence has been modified in the revised manuscript and it now reads: 'Night frosts postponed the street cleaning that commonly starts in March, to the beginning of April.'

*51. P8 L5 Improve sense.*

**Answer:**

The sentence has been removed in the revised manuscript.

*52. P8 L11-14. These correlation stats are of interest. Are sub-daily observed concentration values available? Inspect the average diurnal variation to see if there is any relation to congestion.*

**Answer:**

The R2 values of hourly mean concentrations have been added in the revised manuscript. As expected, correlation on the hourly level is lower compared with the daily level. This can be attributed to differences in diurnal cycle of the modelled and observed concentrations and to the higher uncertainties in predictions of the surface moisture on hourly level.

We have inspected average diurnal cycles of observed and modelled street increments of $PM_{10}$, and traffic volume and speed for years 2009 and 2014. The observed concentrations exhibit more pronounced peak during the evening rush hours, whereas diurnal cycle of the modelled $PM_{10}$ concentrations follow closely the diurnal cycle of the traffic volume and demonstrates peaks of the similar order of magnitude.

*53. P9 L1 May be non-linear due to congestion.*

**Answer:**

The sentence has been modified in the revised manuscript and reads as follows: 'The dependency on vehicle speed may be non-linear for the lower traffic speeds encountered in this study (< 30 km h$^{-1}$) due to congestion.'

*54. P9 L14-19 Doesn't need a table.*

**Answer:**

*Table 6 has been replaced with one sentence of text.*

*55. P9 L29 Compounds or configurations?*

**Answer:**

The correction has been included in the revised manuscript.

*56. P9 last sentence – remove, not of relevance.*

**Answer:**

The sentence has been removed.

*57. P10 L11 remind readers not in FORE.*

**Answer:**

This has been clarified in the revised manuscript.

*58. P10 Don't need Table 8.*

**Answer:**

*Table 8 has been removed.*

*59. P10 around the 2nd paragraph – say something about the consistency of concentrations. If the models predict consistent concentrations why is there a massive difference in predictions in the Feb / March period?*

**Answer:**

We will revise this section.

*60. P10 L19 which met parameters?*

**Answer:**

We refer here to the meteorological conditions in general. This paragraph will be revised.

*61. P10 L37 Us the term hypothetical?*

**Answer:**

Correction has been included in the revised manuscript.

*62. P10 Can some emissions and concentration source apportionment analyses for the road be presented? i.e. in terms of road wear, tyre wear, brake wear, exhaust, resuspension, winter tyres – for both models?*

**Answer:**

Contribution estimation from different sources to the $PM_{10}$ emissions and concentrations, and separation between direct and suspension emissions are possible only by the NORTRIP model. Such results are important; however, we feel that they are beyond the scope of this study.

*63. Section 4, once other revisions to the paper have been completed , Section should be reviewed e.g. P11 L14-15 – the substantial differences need to be made clearer.*

**Answer:**

We will carefully review and revise section 4. We have included a paragraph with comparison of the two road dust emission models as answer to the reviewer's comment #6. This provides also clarification of the substantial differences.

*64. P11 L17 Yes!*

**Answer:**

The sentence will be removed in the revised manuscript.

*65. Table 3 – where does the '5 and 10' referred to in the caption come from? What does the last sentence in the caption mean? How do these values relate to the road? Are any of these values from EMEP?*

**Answer:**

The factor 5 is based on an estimate of the increased number of tyres and studs for a HDV compared to an LDV. The factor of 10 for suspension is taken from other studies where they have derived this value from measurements and regression. The references to this are given in Denby et al. 2013a.

*66. Table 5 (possibly elsewhere) as the models are introduced as NORTRIP then FORE, they should be presented accordingly in the table.*

**Answer:**

Suggested correction has been included in the revised manuscript.

*67. Figure 4 – explain why FORE does badly Jan-Mar i.e. predicts much larger values than NORTRIP for that period.*

**Answer:**

During Jan-March occurrence of the clustered precipitation and dry days was recorded. Dust is accumulated during the wet days and released during the dry days, which is in the model controlled by the reduction factor for moisture content. Large values for this period can be a result of not well reproduced impact of real world surface wetness. From Figure 3 it can be noticed that there are several salting events during Jan-Mar in 2009 and 2014. Application of traction salt can keep surface wet for longer periods. Impact of salt is not considered in the FORE model. Some uncertainty can also occur because the snowfall is treated similarly as rainfall in the model. This is a very good question and extensive analyses would be required for more comprehensive answer.

*68. Reduce the caption length for Figure 5.*
**Answer:**

The caption of the Figure 5 is modified in the revised manuscript as follows: 'Modelled relative changes in the non-exhaust street increment of $PM_{10}$ for analysed cases described in Table 6, averaged over the four years (2007-2009 and 2014). Line markers show values for the individual years.'

References:

Boulter, P. G.: A review of emission factors and models for road vehicle non-exhaust particulate matter, Wokingham, UK, TRL Limited, TRL Report PPR065, URL: http://uk-air.defra.gov.uk/reports/cat15/0706061624_Report1__Review_of_Emission_Factors.PDF, 2005.

Denby, B.R., Sundvor, I., Johansson, C., Pirjola, L., Ketzel, M., Norman, M., Kupiainen, K., Gustafsson, M., Blomqvist, G., Omstedt, G.: A coupled road dust and surface moisture model to predict non-exhaust road traffic induced particle emissions (NORTRIP). Part 1: road dust loading and suspension modelling, Atmospheric Environment, 77, 283-300, 2013a

Denby, B.R., Sundvor, I., Johansson, C., Pirjola, L., Ketzel, M., Norman, M., Kupiainen, K., Gustafsson, M., Blomqvist, G., Kauhaniemi, M., Omstedt, G.: A coupled road dust and surface moisture model to predict non-exhaust road traffic induced particle emissions (NORTRIP). Part 2: Surface moisture and salt impact modelling, Atmospheric Environment, 81, 485-503, 2013b.

Kauhaniemi, M., Kukkonen, J. Härkönen J., Nikmo J., Kangas L., Omstedt G., Ketzel M., Kousa A., Haakana M., Karppinen A.: Evaluation of a road dust suspension model for predicting the concentrations of PM10 in a street canyon, Atmospheric Environment, 45, 3646-3654, 2011.

Kulovuori, S., Ritola, R., Stojiljkovic, A., Kupiainen, K., Malinen, A.: Katupölyn lähteet, päästövähennyskeinot ja ilmanlaatuvaikutukset – Tuloksia KALPA-tutkimushankkeesta 2015–2018. HSY publications 1/2019, in Finnish, summary in Endglish, https://www.hsy.fi/sites/Esitteet/EsitteetKatalogi/Julkaisusarja/1-2019-katupolyn-lahteet-paastovahennyskeinot-ja-ilmanlaatuvaikutukset-KALPA-2015-2018.pdf

Kupiainen, K., Ritola, R., Stojiljkovic, A., Pirjola, L., Malinen, A., Niemi, J.: Contribution of mineral dust sources to street side ambient and suspension PM10 samples, Atmospheric Environment 147, 178-189, 2016.